# Beyond excitation/inhibition imbalance in multidimensional models of neural circuit changes in brain disorders

Cian O'Donnell[1,2]*, J Tiago Gonçalves[3,4], Carlos Portera-Cailliau[4,5], Terrence J Sejnowski[2,6]*

[1]Department of Computer Science, University of Bristol, Bristol, United Kingdom; [2]Howard Hughes Medical Institute, Salk Institute for Biological Studies, La Jolla, United States; [3]Dominick Purpura Department of Neuroscience, Albert Einstein College of Medicine, Bronx, United States; [4]Department of Neurology, David Geffen School of Medicine at UCLA, Los Angeles, United States; [5]Department of Neurobiology, David Geffen School of Medicine at UCLA, Los Angeles, United States; [6]Division of Biological Sciences, University of California at San Diego, La Jolla, United States

**Abstract** A leading theory holds that neurodevelopmental brain disorders arise from imbalances in excitatory and inhibitory (E/I) brain circuitry. However, it is unclear whether this one-dimensional model is rich enough to capture the multiple neural circuit alterations underlying brain disorders. Here, we combined computational simulations with analysis of in vivo two-photon $Ca^{2+}$ imaging data from somatosensory cortex of *Fmr1* knock-out (KO) mice, a model of Fragile-X Syndrome, to test the E/I imbalance theory. We found that: (1) The E/I imbalance model cannot account for joint alterations in the observed neural firing rates and correlations; (2) Neural circuit function is vastly more sensitive to changes in some cellular components over others; (3) The direction of circuit alterations in *Fmr1* KO mice changes across development. These findings suggest that the basic E/I imbalance model should be updated to higher dimensional models that can better capture the multidimensional computational functions of neural circuits.
DOI: https://doi.org/10.7554/eLife.26724.001

**\*For correspondence:**
cian.odonnell@bristol.ac.uk
(CO'D);
terry@salk.edu (TJS)

**Competing interests:** The authors declare that no competing interests exist.

## Introduction

The nervous system shows complex organization at many spatial scales: from genes and molecules, to cells and synapses, to neural circuits. Ultimately, the electrical and chemical signaling at all these levels must give rise to the behavioral and cognitive processes seen at the whole-organism level. When trying to understand prevalent brain disorders such as autism and schizophrenia, a natural question to ask is: where is the most productive level of neuroscientific investigation? Traditionally, most major disorders are diagnosed entirely at the behavioral level, whereas pharmaceutical interventions are targeted at correcting alterations at the molecular level. However, even for the most successful drugs, we have little understanding of how pharmaceutical actions at the molecular level percolate up the organizational ladder to affect behavior and cognition. This classic bottom-up approach may even be further confounded if phenotypic heterogeneity in disorders such as autism turn out not to reflect a unique cellular pathology, but rather 'a perturbation of the network properties that emerge when neurons interact' (*Belmonte et al., 2004*). These considerations imply that a more promising level of analysis might be at the level of neural circuits, since the explanatory gap between circuits and behavior is smaller than the gap between molecules and behavior. This circuit-level viewpoint argues for a reverse-engineering approach to tackling brain disorders: rather than

**eLife digest** In many brain disorders, from autism to schizophrenia, the anatomy of the brain appears remarkably unchanged. This implies that the problem may reside in how neurons communicate with one another. Unfortunately, neuroscientists know little about how brain activity might differ from normal in these disorders, or how specific changes in activity give rise to symptoms. One leading theory, first proposed over a decade ago, is that these disorders reflect an imbalance in the activity of excitatory and inhibitory neurons. Excitatory neurons activate their targets, whereas inhibitory neurons suppress or silence them. While studies in mice have lent support to this theory, they have not yet culminated in new treatments for brain disorders.

One limitation of the excitation-inhibition imbalance theory is that it is one-dimensional. It assumes that there is an optimal balance of excitation and inhibition, and that brain disorders can be arranged in an imaginary line on either side of this optimum. Disorders to the right of the optimum, such as epilepsy and some forms of autism, feature too much excitation. Disorders to the left, such as the developmental disorder Rett syndrome, feature too much inhibition. But can diverse brain disorders really be classified on the basis of a single property, or do scientists need to consider other factors?

To find out, O'Donnell et al. analyzed recordings of brain activity from genetically modified mice with the mutation that causes fragile X syndrome, the most common form of inherited learning disability and autism. The mice showed changes in their overall brain activity compared to control animals. Their neurons also tended to fire in a more synchronized manner. A computer simulation revealed that an imbalance in excitation and inhibition alone could not explain these changes. Yet, a more complex simulation incorporating extra properties of neural circuits did a better job of explaining the altered neural activity seen in the mice.

O'Donnell et al. propose that this more advanced multi-dimensional model of changes in neural circuits could be used to screen candidate drugs before testing them in patients. In principle, the model could even help with designing drugs or other interventions by making it easier for researchers to target more precisely the changes in neural circuits that occur in brain disorders.
DOI: https://doi.org/10.7554/eLife.26724.002

start at the molecular level and working up, we should instead start by asking how cognitive and behavioral symptoms manifest as alterations at the circuit level, then interpret these changes at the levels of cells, synapses, and molecules as appropriate.

One prominent circuit-level hypothesis for brain disorders has been the idea of an imbalance in excitatory and inhibitory signaling. First proposed as a model for autism (*Rubenstein and Merzenich, 2003*), the concept has since been applied to many other brain disorders, including Schizophrenia, Rett syndrome, fragile-X syndrome, tuberous sclerosis, and Angelman Syndrome. However, a major drawback of this model is that it only considers overall activity, which is one-dimensional. It implies that either too much excitation or too much inhibition is unhealthy (*Figure 1A*). Although several studies have found evidence that the E/I balance is indeed upset in multiple brain disorders (*Bateup et al., 2011*; *Dani et al., 2005*; *Gibson et al., 2008*; *Kehrer et al., 2008*; *Wallace et al., 2012*), a model's usefulness should not be judged on whether it is nominally true or false, but on its explanatory and predictive powers as compared with competing alternative models. In this study, we argue that that even if the E/I imbalance model proves correct, its unidimensionality might ultimately limit its applicability, for three reasons.

First, by placing all disorders on the same single axis, the E/I imbalance model implicitly lumps together some vastly different disorders, such as epilepsy, schizophrenia and autism (*Figure 1A*) because they share an excess of excitation. By extension it implies that the symptoms of diverse disorders could be normalized solely by either enhancing or reducing the level of, say, GABAergic signaling as appropriate. Although clinical trials for such GABAergic-based interventions are ongoing (*Braat and Kooy, 2015*), no treatment for a neurodevelopmental disorder based on this principle has yet been approved.

A second issue with the unidimensionality of the E/I imbalance model is that it lumps together all excitatory and inhibitory neural circuit components. In *Figure 1B*, we show a schematic diagram of a

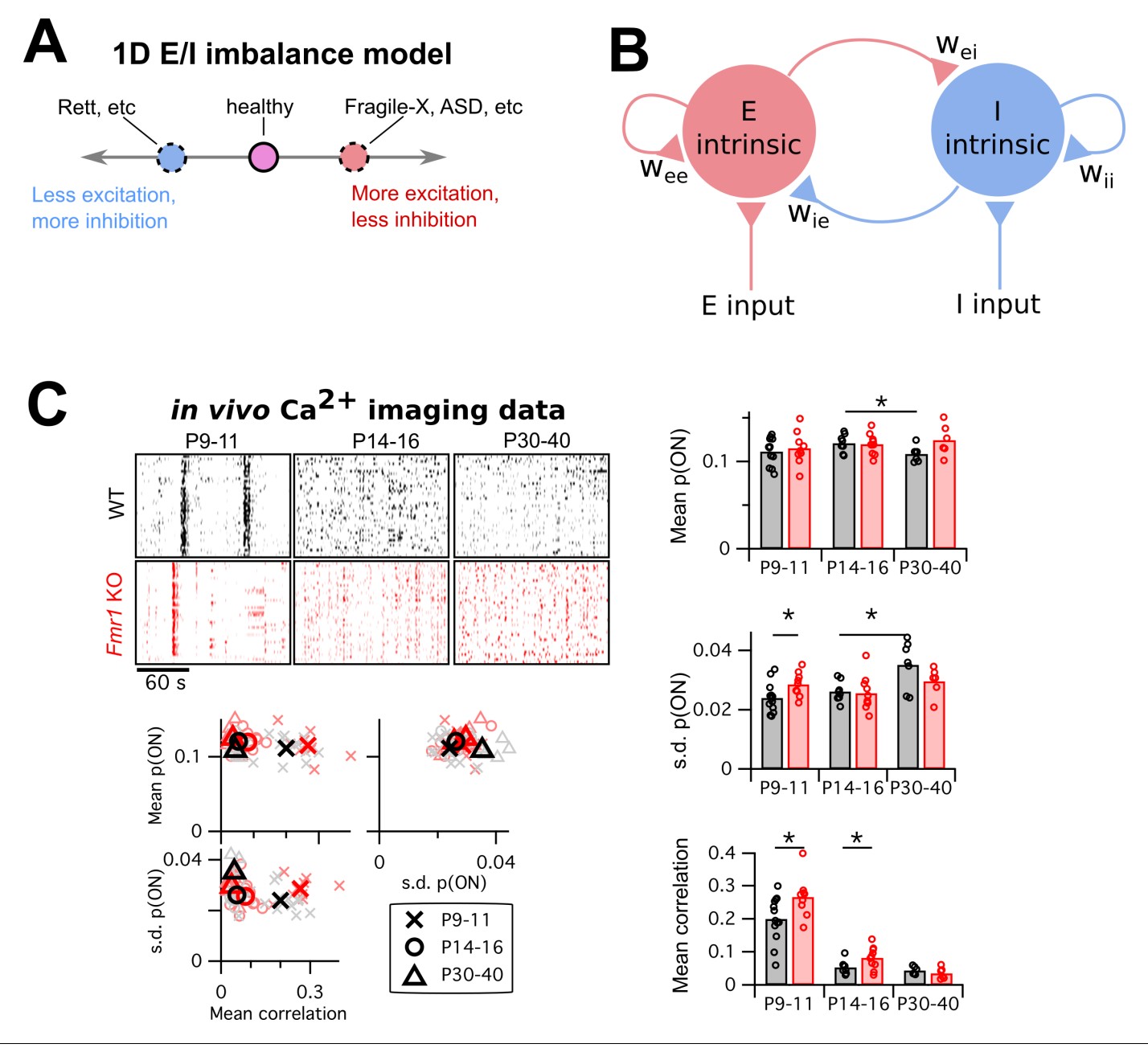

**Figure 1.** Mismatch between the E/I imbalance model's unidimesionality and the multiple changes in circuit activity in Fragile-X mouse models. (A) Schematic of standard E/I imbalance model as a unidimensional axis. Fragile-X Syndrome and autism have been associated with an excess of excitation (**Gibson et al., 2008**; **Lee et al., 2017**; **Nelson and Valakh, 2015**), while Rett Syndrome has been associated with an excess of inhibition (**Dani et al., 2005**). (B) Diagram of a generic neural circuit, showing an excitatory and an inhibitory population of neurons and their interconnections. Although the E/I imbalance model implicitly groups all components as either excitatory (red) or inhibitory (blue), in principle any component could separately be altered in brain disorders, and may have a distinct effect on circuit function. (C) Upper left, example Ca$^{2+}$ imaging dF/F raster plots from a single animal from each of two genotypes, WT and *Fmr1* KO, and three age groups, P9–11, P14–16 and P30–40. In each case, 3 min of data are shown from 40 neurons. Right and lower left, mean firing probability, standard deviation of firing probabilities, and mean pairwise correlation across all neurons. Same data in scatter plots lower left and bar charts right. * indicates significant difference in group means at $p < 0.05$, by bootstrapping.
DOI: https://doi.org/10.7554/eLife.26724.003

generic neural circuit with excitatory components colored red and inhibitory components colored blue. The E/I imbalance model implies that varying any of the excitatory components, such as the strength of recurrent excitatory synapses or the input resistances of excitatory neurons, would have the same overall effect on circuit function. In contrast, theorists have found that these equivalences often do not hold even in very simple circuit models (*Wilson and Cowan, 1972*).

Third, because the standard E/I imbalance model is given in terms of circuit components, not circuit function, it does not specify which aspect of a neural circuit's activity should be maintained for healthy performance. For example, it leaves unclear which of neuronal firing rates, synchrony, or reliability of responses might be altered if E/I balance is upset.

To motivate our study, we began by investigating which circuit activity properties are altered in a model brain disorder. We re-analyzed published in vivo two-photon $Ca^{2+}$ imaging data we previously recorded from somatosensory cortex in *Fmr1* knockout mice (*Gonçalves et al., 2013*), a well-studied animal model for fragile-X syndrome (*The Dutch-Belgian Fragile X Consortium, 1994*). We compared the data from wild-type (WT) mice with *Fmr1* KO mice, across three different developmental time points: just before (P9–11) and after (P14–16) the critical period for heightened activity-dependent synaptic plasticity in L2/3 barrel cortex, and a more mature timepoint (P30–40). Example $\Delta F/F$ raster plots from each group are shown in *Figure 1C*, top left. We binned the data into 1 s timebins (originally imaged at 4 Hz), then transformed each neuron's timeseries of $\Delta F/F$ values into a probabilistic sequence of binary ON/OFF values by assuming a Poisson firing model (Materials and methods). We then summarized the neural population activity from each animal with three statistics: the mean ON probability across all recorded neurons, the standard deviation (s.d.) in ON probability across neurons, and the mean correlation between all pairs of neurons (*Figure 1C*, bar charts right and scatter plots lower left). Together these measures capture both the statistics of the bulk population activity and some indication of the heterogeneity across neurons.

For mean firing rates, the only change we detected was a decrease in firing probability in WT between P14–16 and P30–40 (p=0.027), which was coupled with an increased s.d. of firing rates (p=0.015). We also detected a higher firing rate s.d. in P9–11 KO animals than WT (p=0.031). Finally, as previously reported (*Golshani et al., 2009*; *Gonçalves et al., 2013*; *Rochefort et al., 2009*), we found a substantial decrease in pairwise correlations in both genotypes across development, with slightly higher correlations in KO animals than WT at P9–11 (p=0.029) and P14–16 (p=0.047).

These results show that multiple statistics of cortical circuit activity are altered in *Fmr1* KO mice. However, two questions remain: (1) Which circuit components are responsible for these activity alterations? (2) What aspects of these activity alterations impact circuit computation? In the remainder of this study, we used computational simulations and further data analysis to ask whether the E/I imbalance model could help address these questions.

We first built a detailed spiking neural circuit model of mouse L2/3 somatosensory cortex to explore how its various excitatory and inhibitory components affect the circuit's spiking output, and found that the E/I imbalance model was not flexible enough to capture key aspects of this relationship. We then derived an abstract two-dimensional circuit model that captures more features of the circuit function than the 1-dimensional E/I imbalance model did. Using this new model, we found that certain sets of circuit components have redundant effects on circuit function, and that circuit function is vastly more sensitive to changes in some components over others. To ask how this 2D model could help interpret brain circuit abnormalities in a particular test case, we fit a version of the model to the $Ca^{2+}$ imaging data from Fragile-X mouse models presented above. We found that the model predicts opposite changes in *Fmr1* KO circuit properties at different developmental ages. Finally, we applied a new large-scale neural population analysis method (*O'Donnell, 2017b*) to the same $Ca^{2+}$ imaging data, and found systematic shifts in the distribution of neural circuit activity patterns in Fragile-X that were not predictable from neural firing rates or correlations alone.

## Results

Neural circuits consist of many components that typically interact non-linearly to generate complex circuit activity dynamics. Although many properties of cortical circuit components have been found to be altered in animal models of brain disorders, it remains extremely difficult to predict the net effect of varying any one particular parameter on circuit activity. The E/I imbalance model seeks to simplify this problem by projecting all circuit alterations onto a one-dimensional axis (*Figure 1A*)

where the goal is to achieve a 'healthy' balance of excitation and inhibition. Under this model, either too much excitation or too much inhibition leads to improper circuit function.

To explicitly test whether the E/I imbalance model can account for the effects of cellular component changes on circuit function, we built a detailed computational model of layer (L) 2/3 mouse somatosensory cortex. This circuit has been studied in detail by neurophysiologists, and several of its properties have been found to be altered during development in mouse models of Fragile-X syndrome, including parvalbumin-positive interneuron number (*Selby et al., 2007*), GABA receptor reversal potential (*He et al., 2014*), dendritic spine dynamics (*Cruz-Martín et al., 2010*), and L4 excitatory input (*Bureau et al., 2008*), reviewed by *Contractor et al., 2015*. We used numerical computer simulations to perform hypothetical experiments where we perturbed various parameters of the circuit model and observed the resulting changes in circuit-level activity. Although we focused on this particular brain circuit for tractability, our general conclusions and methodology should be readily applicable to other brain circuits (*Frye and MacLean, 2016*).

The L2/3 computational model we built (*Figure 2A*, see Materials and methods for details) consisted of four interconnected populations of leaky integrate-and-fire neurons: one group of 1700 excitatory (E) pyramidal neurons and three groups of inhibitory neurons: 115 $5HT_{3A}R$-expressing neurons, 70 parvalbumin-expressing (PV) neurons, and 45 somatostatin-expressing (SOM) neurons. This L2/3 circuit was driven by a separate population of 1500 L4 excitatory neurons. Cellular numbers, intrinsic properties, synaptic strengths, and connectivity statistics were taken from published in vitro data from P17–22 wild-type mice (*Avermann et al., 2012*; *Lefort et al., 2009*; *Tomm et al., 2014*). We chose this level of detail for the model in order to relate experimentally measureable biophysical properties of neurons to their putative role in the circuit at large.

L2/3 neurons of the rodent somatosensory neocortex respond only sparsely to sensory stimulation in vivo. For example, twitching a whisker activates, on average, only ~20% of L2/3 neurons in its corresponding barrel, each of which typically emits only one action potential (*Clancy et al., 2015*; *Kerr et al., 2007*; *Sato et al., 2007*). Hence any individual neuron carries very little information about the stimulus on its own, implying that information must instead be encoded at the circuit level as the identities of the subset of neurons that respond.

To model whisker stimulation, we simulated a volley of spikes arriving from L4 as input to the population of L2/3 cells. We chose a random subset of L4 neurons as ON, then sent a single spike from each of these L4 cells to their target neurons in L2/3, and recorded the responses of all neurons in L2/3, some of which spiked and some of which did not. We repeated this identical stimulation multiple times, in order to get an average response probability for each L2/3 neuron, given the probabilistic vesicle release at synapses in the model. Then, we chose a different random subset of L4 neurons as ON, and repeated the entire procedure. Finally, we varied the fraction of L4 cells active and plotted the probability of response for each individual L2/3 neuron as a function of L4 activity level (*Figure 2B–C*).

We found that the mean response probability of each neuron in the simulation increased from zero to one monotonically with increasing L4 activity level. This sigmoidal-shaped response profile of simulated L2/3 neurons mimics the spiking response of mouse L2/3 pyramidal cells to extracellular L4 stimulation in vitro (*Elstrott et al., 2014*), while the sparse, noisy and distributed network responses were reminiscent of in vivo activity following whisker stimulation (*Clancy et al., 2015*; *Kerr et al., 2007*). Neurons of all four cell types in the simulation responded to the L4 stimulus, including the SOM interneurons which did not receive direct L4 input, but were instead activated by disynaptic connections via L2/3 excitatory neurons. The detailed shape of the response curve varied systematically across cell types and was heterogeneous for different neurons of a given cell type. To quantify these differences, we used logistic regression to fit the response profile of each neuron with a sigmoid function (*Figure 2D*, inset), which has just two parameters: the slope (representing the steepness of the response curve) and threshold (representing the minimal fraction of L4 neurons needed to activate the cell). When we plotted the fitted slope and threshold values for each neuron against each other, we found that each cell type falls into a distinct cluster in this two-dimensional space. For example, all PV inhibitory neurons had a low slope and low threshold, whereas SOM inhibitory neurons had a steep slope and moderate threshold. We then used these slope-threshold measurements to summarize the circuit-level input-output function of this 'default' model of L2/3 somatosensory cortex. This 2D logistic model has two benefits over the 1-D E/I imbalance model: first, its extra degree of freedom allows for richer and more flexible fits to data, and second, by

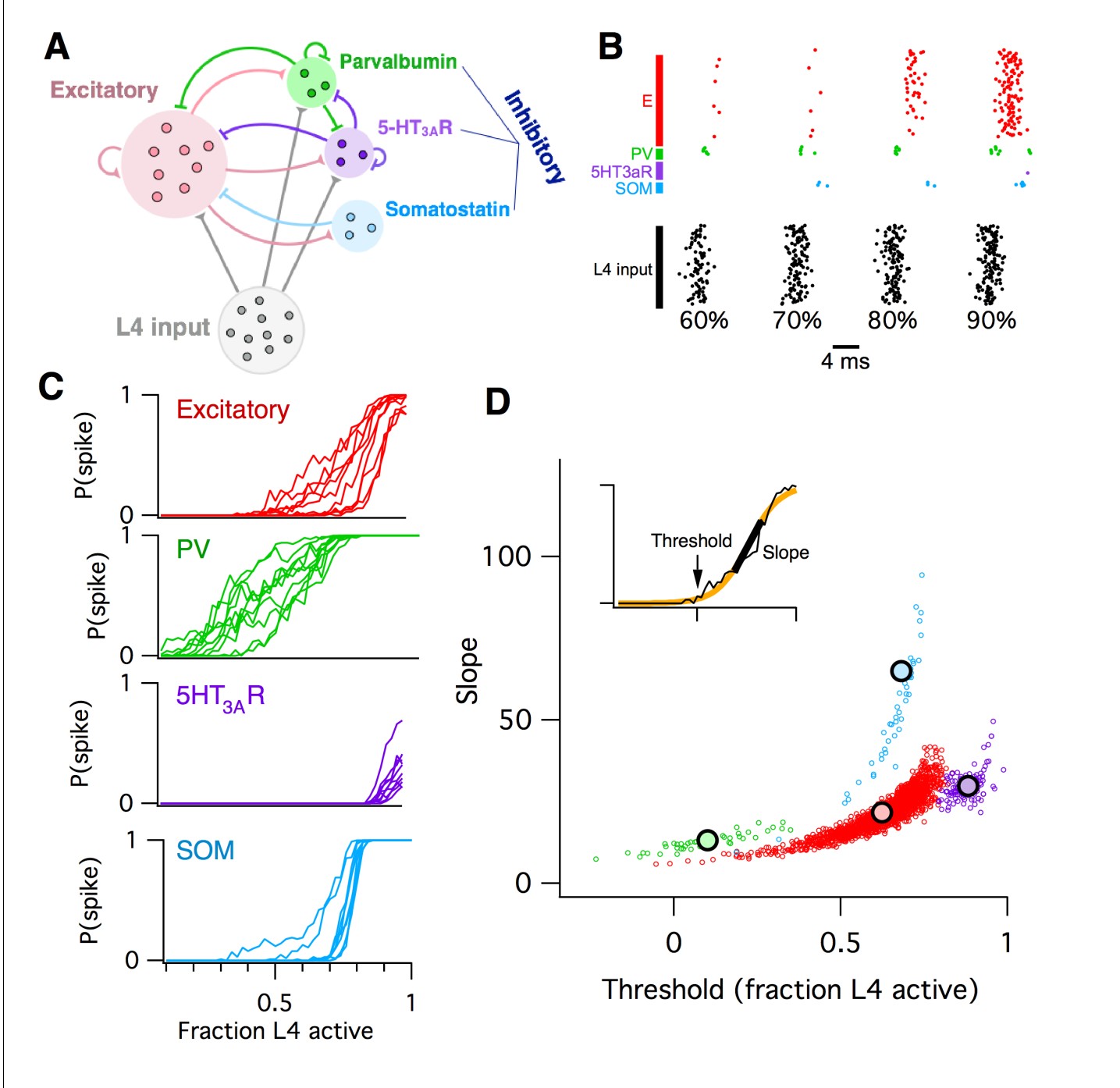

**Figure 2.** Computational model of L2/3 mouse somatosensory cortex. (A) Schematic diagram of computational circuit model. (B) Example raster plots of spiking responses from a subset of neurons from each cell type (colors as in panel A), for varying fractions of L4 activated (black). (C) Probability of spiking as a function of the fraction of L4 neurons activated. Each curve represents the response probability of a single neuron, averaged over multiple trials and multiple permutations of active L4 cells. (D) Each circle plots the fitted logistic slope and threshold values for a single neuron in the simulation. Circle color indicates cell type: red is excitatory, green is PV inhibitory, purple is $5HT_{3A}R$ inhibitory, blue is SOM inhibitory. Large black circles indicate mean for each cell-type. Inset shows an example fitted logistic response function (orange) to the noisy simulation results from a single excitatory neuron (black).

DOI: https://doi.org/10.7554/eLife.26724.004

describing an input-output mapping for the L2/3 circuit it can capture some aspects of the computation that the circuit performs for the animal. In contrast, the E/I imbalance model is specified purely in terms of circuit components, and so is agnostic to the circuit's computational function.

The biophysical circuit model contained 100 parameters (Materials and methods). How sensitive is the circuit's macroscopic input-output function to alterations in its low-level components? To test this, we varied 76 of the model parameters in turn by ±20%, and repeated the entire set of simulations for each case (Materials and methods). For each individual parameter alteration, we fit a new logistic response function for each model neuron. We summarize the results by plotting the logistic slope and threshold parameters and comparing their values to those found with the default model. The outcomes were hugely varied. We show three examples from the set of 76 in *Figure 3A*, chosen to illustrate three qualitatively different effects that neural parameter changes can have on circuit function. First, when we increased the amplitude of postsynaptic potentials (PSPs) of excitatory synapses from L4 to L2/3 excitatory neurons, we found that the logistic threshold parameter of all cell types shifted leftwards to lower values (*Figure 3A* left), implying that fewer L4 neurons were needed to activate the entire L2/3 circuit. In contrast, when we increased the PSP amplitude of a different excitatory synapse, the recurrent connections between L2/3 excitatory neurons, we found (*Figure 3A* center) that excitatory and SOM inhibitory neurons had increased slope parameters relative to default, with little change in their threshold parameters. At the same time, 5HT$_{3A}$R inhibitory neurons had decreased slopes and thresholds, while PV neurons had little change at all. As a third example, we increased the probability of inhibitory synaptic connections from L2/3 PV interneurons to L2/3 excitatory neurons (*Figure 3A* right). In this case, we found that excitatory neurons had a marginally lower slope and increased threshold, SOM inhibitory neurons had a lower slope, 5HT$_{3A}$R inhibitory neurons had both an increased slope and threshold, and PV neurons showed little change, even though their outgoing synapses were the parameter that was altered.

To synthesize the findings from all simulations, for each altered parameter we plotted the shift in mean slope-threshold values for L2/3 excitatory neurons from the mean values found with the default model (*Figure 3B*). We focused on excitatory neurons because they constitute 90% of the neurons in this layer (*Lefort et al., 2009*) and are the primary output to downstream circuits (*Mao et al., 2011*; *Petreanu et al., 2007*). Overall, we found a very heterogeneous picture. First, the magnitude of the shift in circuit response varied from parameter to parameter (*Figure 3B–C*). Varying some parameters, such as the first two examples given above, had large effects, whereas varying other parameters such as $w_{I5htE}$ (the strength of synapses from 5HT$_{3A}$R inhibitory neurons to E neurons) or $\tau m_{Isom}$ (the membrane time constant of SOM inhibitory neurons) had little effect. Second, the direction of shift in 2-D slope-threshold space also depended on parameter (*Figure 3C*). Increasing some parameters changed either circuit slope or threshold in isolation, while other parameters changed both slope and threshold together. All four quadrants of the slope-threshold plane could be reached by various subsets of the model parameters. Third, basic knowledge of whether a component was 'excitatory' or 'inhibitory' was insufficient to predict the direction of slope-threshold change. For example, the two glutamatergic projections considered in *Figure 3A* had distinct effects on circuit function.

In summary, these simulations indicate that the L2/3 somatosensory cortex circuit has extremely varied sensitivities to changes in its cellular components and that the eventual circuit-level consequences cannot be predicted from knowledge of the class of the perturbed neurotransmitter alone. Since the E/I imbalance model groups all excitatory and inhibitory components as respective equals, it cannot account for these results.

## Firing rates and correlations from the logistic model

In the above analysis, we investigated how low-level circuit components affect a high-level circuit input-output function, as parameterized by the slope and threshold of fitted logistic functions. But how is this logistic input-output function related to more common measures of neural population activity, such as firing rates and pairwise correlations between neurons? To investigate this, we considered the following reduced statistical model of cortical activity. We assumed for simplicity that the magnitude of the total input to the L2/3 circuit can be described by a Gaussian distributed random variable, with zero mean and unit standard deviation (*Figure 4A* lower left). Then we described each L2/3 neuron's input-output as a logistic function as before (*Figure 4A* upper left), with threshold and slope defined relative to the Gaussian input's mean and standard deviation, respectively.

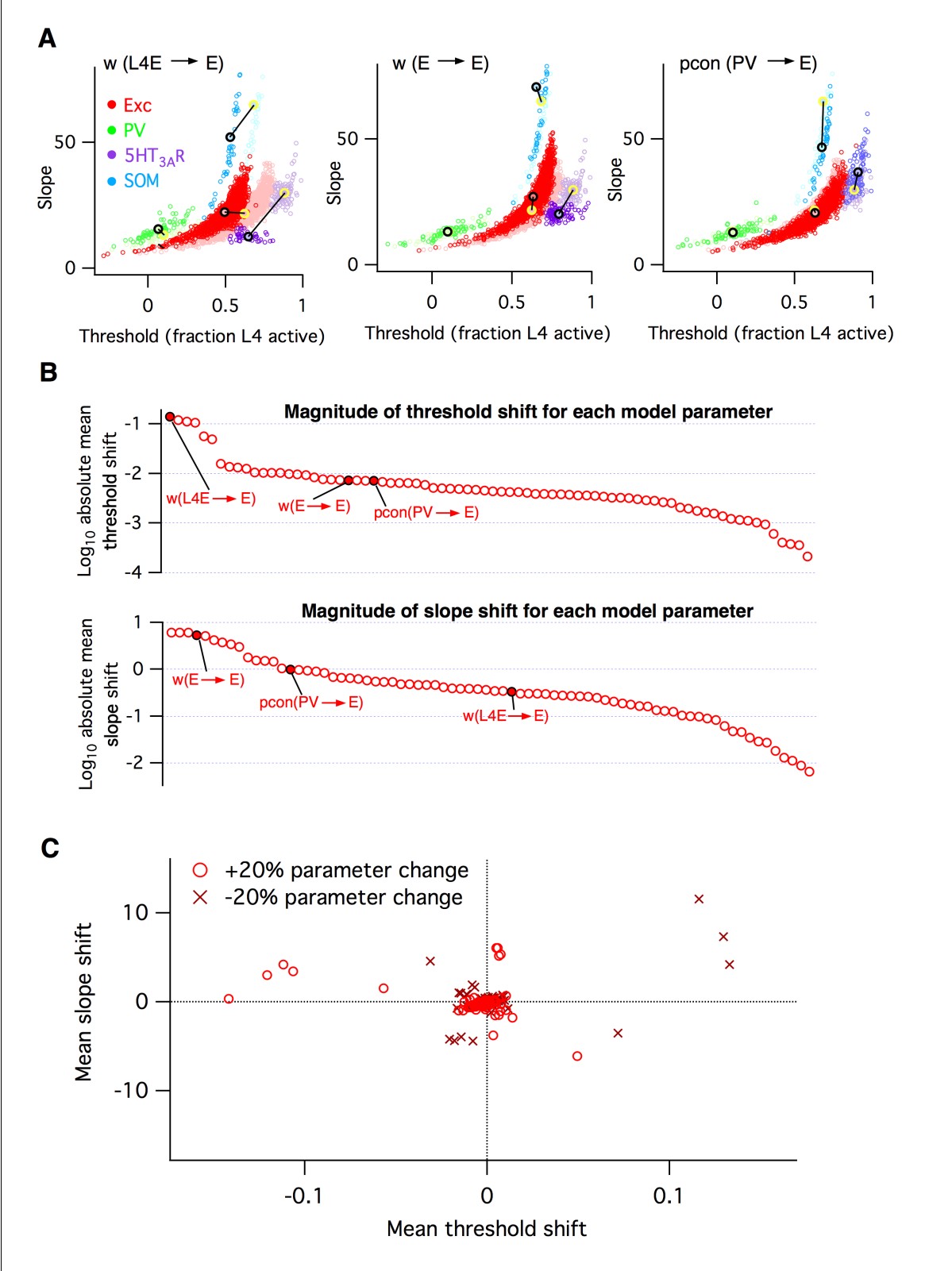

**Figure 3.** Heterogeneous effects of varying L2/3 parameters on the circuit input-output function. (A) Shifts in the distribution of fitted slope and threshold parameters as a result of increasing the strength of synapses from L4 to L2/3 E neurons (left), increasing the strength of recurrent synapses between L2/3 E neurons (center), or increasing the connection probability between L2/3 PV interneurons and E neurons (right). Transparent circles represent values for default network, heavy circles for altered network. The default and altered group means are large yellow and black open circles,

*Figure 3 continued on next page*

*Figure 3 continued*
respectively. (B) Absolute values of mean shifts in threshold (upper plot) and slope (lower plot) for Excitatory neurons arising from increasing the value of each parameter by +20%. The three example parameters from panel A are labeled and indicated as filled red circles. Note that data are presented on a log10 scale. (C) The shift in mean slope-threshold parameter values for E neurons from the default network values, in response to each of the 76 circuit parameter alterations. Light red circles indicate +20% increase in parameter value; dark red crosses indicate a −20% decrease in parameter value.
DOI: https://doi.org/10.7554/eLife.26724.005

Given this model, we can numerically calculate the probability distribution over a neuron's firing probability, which in general is skewed and non-Gaussian (*Figure 4A* upper right). From this function, we compute (Materials and methods) both the neuron's mean firing probability and the pairwise correlation of two identical neurons following this profile (*Figure 4A* lower right). Example samples from the model are illustrated in *Figure 4B*.

Neural firing rates and correlations had qualitatively different dependencies on the underlying logistic model's slope and threshold. Neural firing rate was greatest when threshold was low and slope was high (top left of phase plot, *Figure 4C* left), whereas correlations were greatest when both threshold and slope were high (top right of phase plot, *Figure 4C* right). This implies that any change in the circuit's input-output function slope or threshold will in general have distinct effects on firing rate versus correlations, and so could not be captured by a 1-dimensional E/I balance model that sought to, for example, normalize firing rates alone. To illustrate this fact, we plot the calculated correlation values along a contour where firing probability is fixed at 0.1 (*Figure 4D*). In the region of parameter space where both the slope and threshold are low (*Figure 4C* bottom left), correlations are low,~0.01. However, as we move along the contour for firing rate = 0.1 toward the region of parameter space where slope and threshold are high (*Figure 4C* top right), the pairwise correlations increase to ~0.4. This shows that a one-dimensional E/I balance rule that exclusively sought to normalize neural firing rates would leave neural correlations free to achieve arbitrary values.

Previous studies have found evidence for an E/I imbalance in ASD (*Lee et al., 2017*; *Nelson and Valakh, 2015*). Fragile-X syndrome is the leading inherited cause of ASD, and also carries alterations in excitability (*Contractor et al., 2015*). We aimed to interpret our *Fmr1* KO $Ca^{2+}$ imaging data (*Figure 1C*) via the 2D logistic model. Since our earlier analysis found substantial within-animal heterogeneity in neural activity levels in both genotypes (*Figure 1C*), we extended the 2D logistic model for single neurons to a 5D neural population version that captured cell-to-cell heterogeneity. The three extra parameters represented the standard deviations and correlation in slope and threshold parameters across the neural population (see Materials and methods for details). We fit the parameters of the logistic model to reproduce the same neural population $Ca^{2+}$ imaging data presented in *Figure 1C*. Given the three summary statistics from each animal (firing rate mean and s.d., and mean pairwise correlation), we used a gradient descent algorithm to find the five parameters of the population-level version of the logistic model that best matched the activity statistics (see Materials and methods). The output statistics of the fitted models matched well those of the target data (*Figure 5–figure supplement 1*). Example neural population activity patterns drawn from the mean model fits for each group are shown in *Figure 5A*, along with the fitted slope-threshold functions (*Figure 5A* insets), to be compared with the $Ca^{2+}$ imaging rasters in *Figure 1C*. We also plot the full 5D parameter fits for all animals in *Figure 5–figure supplement 2*. For the rest of the analysis, we focus on the mean slope and mean threshold parameters, which showed the most prominent changes. In *Figure 5B*, we plot the mean slope and mean threshold fits on top of the previously calculated (*Figure 4C*) 2D slope-threshold maps of firing rate and correlation. We found that in young animals, P9–11, most points were scattered at high values of both slope and threshold (*Figure 5B* left). With age, the parameter fits for both genotypes moved south-west toward the low slope and low threshold region of parameter space (*Figure 5B* center and right). The mean location of the cloud of points at each developmental age differed between WT and KO. We plot the direction of shift in group mean from WT to KO in *Figure 5C*. In young animals, P9–11 and P14–16, the KO group had both higher slope and higher threshold than WT, whereas in adult animals, P30–40, the KO group had a lower slope and lower threshold than WT. These results demonstrate an opposite direction of circuit parameter change in young Fragile-X mice compared to adults, which was not be uncovered by measures of neural firing rates and correlations (*Figure 1C*).

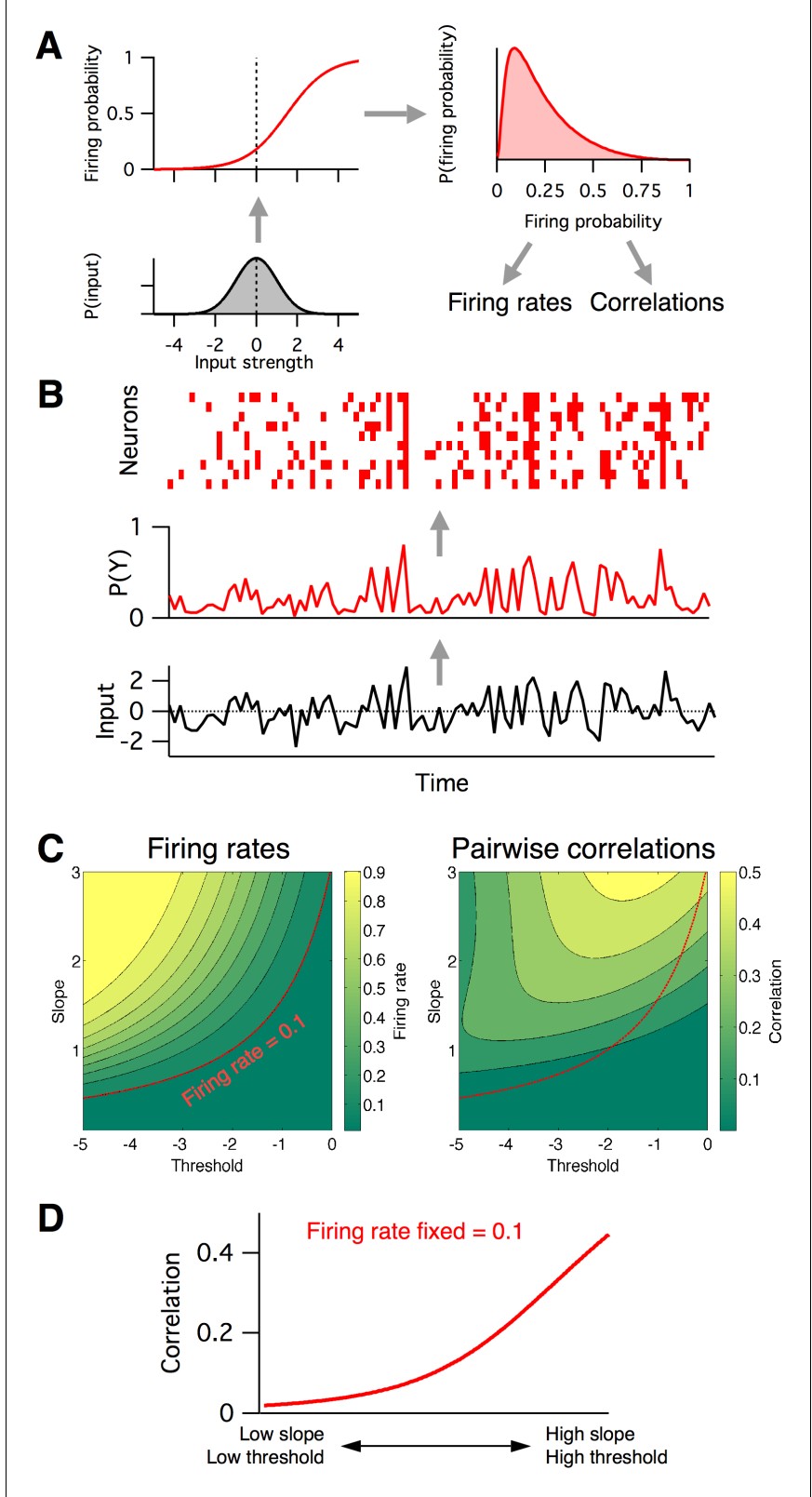

**Figure 4.** Firing rates and pairwise correlations from the logistic response model. (A) Logistic model components. We assume a normally distributed input drive (gray distribution, bottom left), which is passed through the neuron's probabilistic spike input-output function (red curve, top left), which results in a distribution of spike probabilities (top right) that are determined by the input-output function's slope and threshold parameters. From the output

*Figure 4 continued on next page*

*Figure 4 continued*

distribution, we can directly calculate the mean firing rate and correlation between a pair of such neurons (Materials and methods). (**B**) Example spikes from the logistic model. The bottom trace (black) shows examples inputs over time drawn randomly from the same normal distribution. This is transformed to spike probability at each time point (red trace). Example spike trains can then be generated from the spike probability trace by drawing Bernoulli samples with the specified probabilities (red ticks, top). If each neuron's spike train is conditionally independent given the same spike probabilities, we can see correlations in their spike trains. (**C**) Calculated mean firing rate (left) and pairwise correlation (right) color maps as a function of the logistic threshold (x-axis) and slope (y-axis) parameters. Contours indicate lines of fixed firing rate or correlation in the 2D slope-threshold space. (**D**) Pairwise correlation values along the slope-threshold contour for firing rate = 0.1.

DOI: https://doi.org/10.7554/eLife.26724.006

Earlier we asked how sensitive the logistic model slope and threshold parameters were to altera-tions in the many underlying neural circuit components (*Figure 3*). In a similar way, we can also ask how sensitive the neural firing rates and correlations are to alterations in the logistic slope and threshold parameters. This is important since inspection of the two-dimensional maps in *Figure 3C* shows that these sensitivities will differ depending on starting location within the slope-threshold

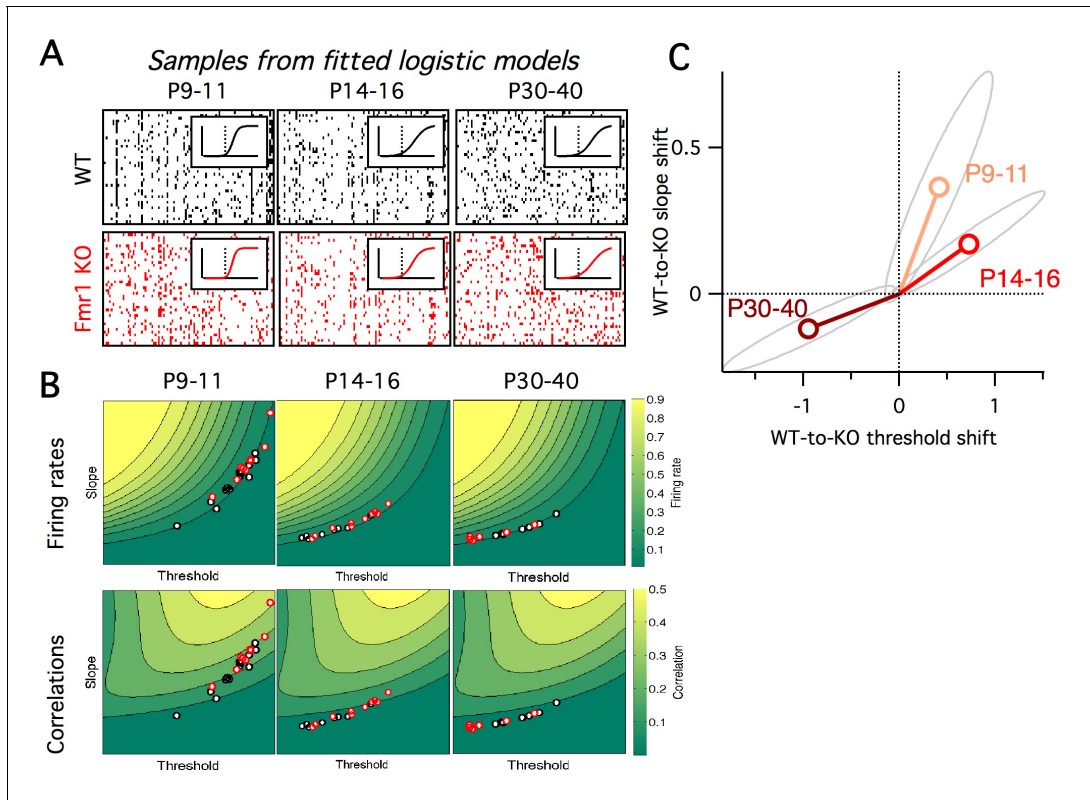

**Figure 5.** Fragile-X fits from logistic model. (**A**) Example samples from the fitted logistic models, corresponding to the six groups shown in panel A. Inset shows group mean fitted logistic function, dashed vertical line represents zero. (**B**) Fitted logistic mean slope and mean threshold values for data from each WT (black circles) and *Fmr1* KO (red circles) animal. Values overlaid on same firing rate (top) and correlation (bottom) maps from *Figure 4C*. (**C**) Shift in mean logistic slope and threshold values from WT to KO for P9–11 (orange), P14–16 (red) and P30–40 (brown). Grey ellipses represent 95% confidence intervals (Materials and methods).

DOI: https://doi.org/10.7554/eLife.26724.007

The following figure supplements are available for figure 5:

**Figure 5—figure supplement 1.** Agreement between population logistic model activity statistics and raw data statistics.

DOI: https://doi.org/10.7554/eLife.26724.008

**Figure 5—figure supplement 2.** Variation in population logistic model parameter fits with developmental age group and genotype.

DOI: https://doi.org/10.7554/eLife.26724.009

space. To quantify this effect, we calculated the sensitivity of both the firing rate and correlations to small changes in the slope and threshold (*Figure 6A–B*, see Materials and methods), quantified as the partial derivatives local to the fitted logistic parameter values for each animal (black and red circles in *Figure 5B*). In general, increasing the slope or decreasing the threshold always increased both firing rates and correlations, as can be predicted from *Figure 5B*. However, the magnitude of sensitivities varied across animals. We found only minor differences in sensitivities between geno-types (*Figure 6C–D*), and as a result we pooled the sensitivity measurements across genotypes to test for statistical differences in sensitivity with developmental age. In young animals, P9–11, changes in the logistic threshold (solid bars in *Figure 6*) had substantial effects on both firing rates and correlation. This sensitivity decreased with age ($p \leq 0.013$ for firing rates, $p < 0.01$ for correlations from P9–11 to P14–16), so that in adult animals, P30-40, changes in threshold had relatively little effect on neural activity statistics. A different picture emerged for the logistic slope parameter (striped bars in *Figure 6*). There, the firing rate sensitivity increased from P9–11 to P14–16 ($p < 1e\text{-}6$) (*Figure 6C*), while correlation sensitivity stayed approximately constant ($p \geq 0.18$) (*Figure 6D*). These results show that the quantitative relationships between neural activity statistics and the parameters of the logistic model, and perhaps also the underlying circuit components, are not fixed across development.

What are the functional implications of these alterations in firing rates and correlations in Frag-ile-X mice across development? To address this, we calculated the entropy of the neural population activity for the data from each animal. Entropy is a quantity from information theory, measured in bits, that puts a hard upper bound on the amount of information that can be represented by any coding system (*Cover and Thomas, 2006*). Intuitively, the entropy measures how uniform the neural population activity pattern distribution is: it is large if the circuit exhibits many different activity pat-terns over time, and small if only a few activity patterns dominate. Entropy is an appealing measure for the present problem because it is sensitive both to neural firing rates and to correlations at all orders. It is typically highest when firing rates are high and correlations are low. Although entropy is notoriously difficult to calculate for large neural populations because most estimation methods require impractically long data recordings (*Quian Quiroga and Panzeri, 2009*), we recently devel-oped a new statistical method for this purpose, called the *population tracking model*, that scales well to large numbers of neurons, even for limited data (*O'Donnell, 2017b*). This model matches both the synchrony distribution for the number of neurons simultaneously active, and the variations in individual cell-to-cell firing rates. We fit this population tracking model to the same $Ca^{2+}$ imaging data as analyzed above (*Figure 7*). An intermediate step in estimating the neural entropy involves calculating a low-parameter approximation of the entire probability distribution over all $2^N$ neural population activity patterns, where $N$ is the number of neurons. The cumulatives of these probability distributions calculated for 50-neuron subsets of the recordings are shown in *Figure 7A*. In young animals P9–11, a small number of activity patterns accounts for a large fraction of the probability mass (*Figure 7A* left). For example, based on these curves, 50% of the time we would expect to see the same 1000–10,000 patterns out of a possible total $2^{50} \approx 10^{15}$ patterns. In contrast, in older ani-mals P14–16 and P30–40 the cumulative distributions shift rightwards so that more patterns are typi-cally observed (*Figure 7A* center, right). In these cases, around 1,000,000 patterns are needed to account for 50% probability mass.

Instead of attempting to quantify these shifts by asking how many patterns are needed to cross an arbitrary threshold of probability mass, we instead calculated the entropy, which takes into account the shape of the entire probability distribution. The entropy depends on the number of neu-rons analyzed, so we normalized all estimates to calculate the entropy per neuron (*Figure 7B–C*). Since we are treating neurons as binary, the entropy/neuron was bounded between 0 and 1 bits. For all age groups, and for both WT and *Fmr1* KO animals, entropy/neuron progressively decreased with the number of neurons analyzed (*Figure 7B*). Because each imaging session captured a differ-ent number of neurons (range 40–198, median 97), we fit the entropy/neuron versus number of neu-rons data with a double exponential function (see Materials and methods) and use the fit to provide a standardized estimate of the entropy/neuron for 100-neuron populations (*Figure 7C*). In WT ani-mals, entropy/neuron showed a non-monotonic trajectory across development (*O'Donnell, 2017b*). At P9–11 it was low, 0.38 bits (95% c.i. [0.35:0.41]), before increasing at P14–16 ($p < 0.001$) to 0.50 bits (95% c.i. [0.48:0.52]), before decreasing again at P30–40 ($p = 0.028$) to 0.45 bits (95% c.i. [0.42:0.48]). We found a different entropy trajectory in *Fmr1* KO animals. There, although entropy/

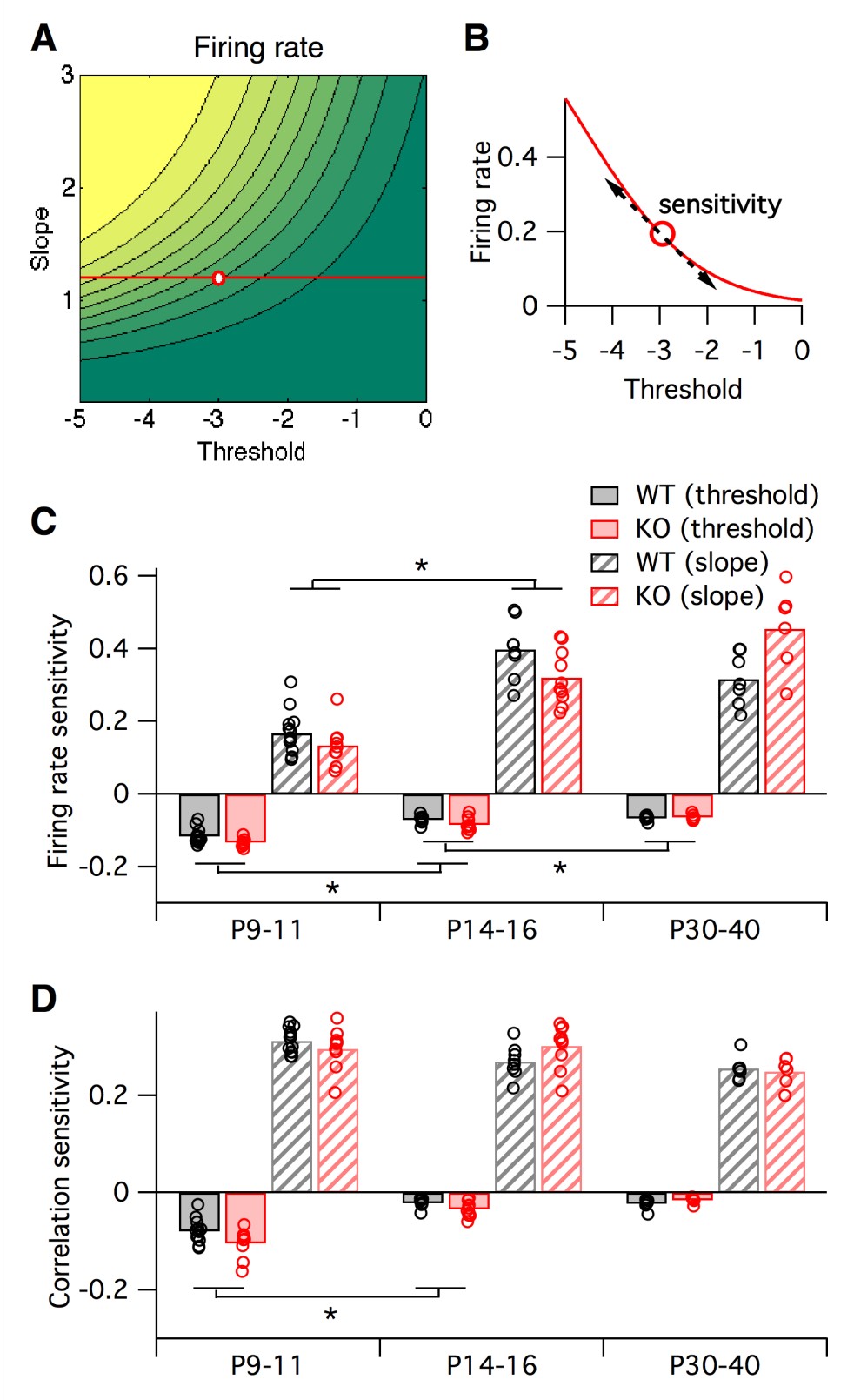

**Figure 6.** Sensitivity of firing rate and correlations with respect to logistic model parameters, local to the parameter fit for each animal. (A–B) Sensitivity to a parameter is calculated about a given point in parameter space. In this hypothetical example, we plot a slope-threshold parameter fit at the red circle on the firing rate contour map (A). The firing rate varies non-linearly if the threshold is varied away from this point (B). Sensitivity is

*Figure 6 continued on next page*

*Figure 6 continued*
calculated as the local derivative, or slope of the tangent, about the target point. (**C–D**) Sensitivity of firing probability (**C**) and pairwise correlations (**D**) to change in threshold (solid bars) and slope (striped bars) parameters of logistic model, about the fitted parameter values for each animal (circles) displayed in *Figure 5B*. Bars represent group means. Each statistical test compares the mean values between adjacent pairs of age groups, where the data were pooled between genotypes.
DOI: https://doi.org/10.7554/eLife.26724.010

neuron also began low at 0.34 bits (95% c.i. [0.30:0.39]), not different from WT (p=0.19), when it increased at P14–16 (p<0.001) to 0.465 bits (95% c.i. [0.45:0.48]) it remained lower than for WT (p=0.048). Finally, instead of decreasing as in the WT case, entropy continued to increase in P30–40 *Fmr1* KO animals (p=0.033) to 0.51 bits (95% c.i. [0.47:0.55]), higher than WT (p=0.034). These entropy values estimated directly from Ca$^{2+}$ imaging data agreed well with entropy estimates for synthetic data sampled from the previously fit logistic models (*Figure 7–figure supplement 1*). In summary, unlike WT animals, *Fmr1* KO mice showed a monotonically increasing entropy/neuron from P9–11 to P30–40. Furthermore, the direction of change in entropy between P14–16 and P30–40 was opposite for WT and *Fmr1* KO animals, decreasing in the former and increasing in the latter.

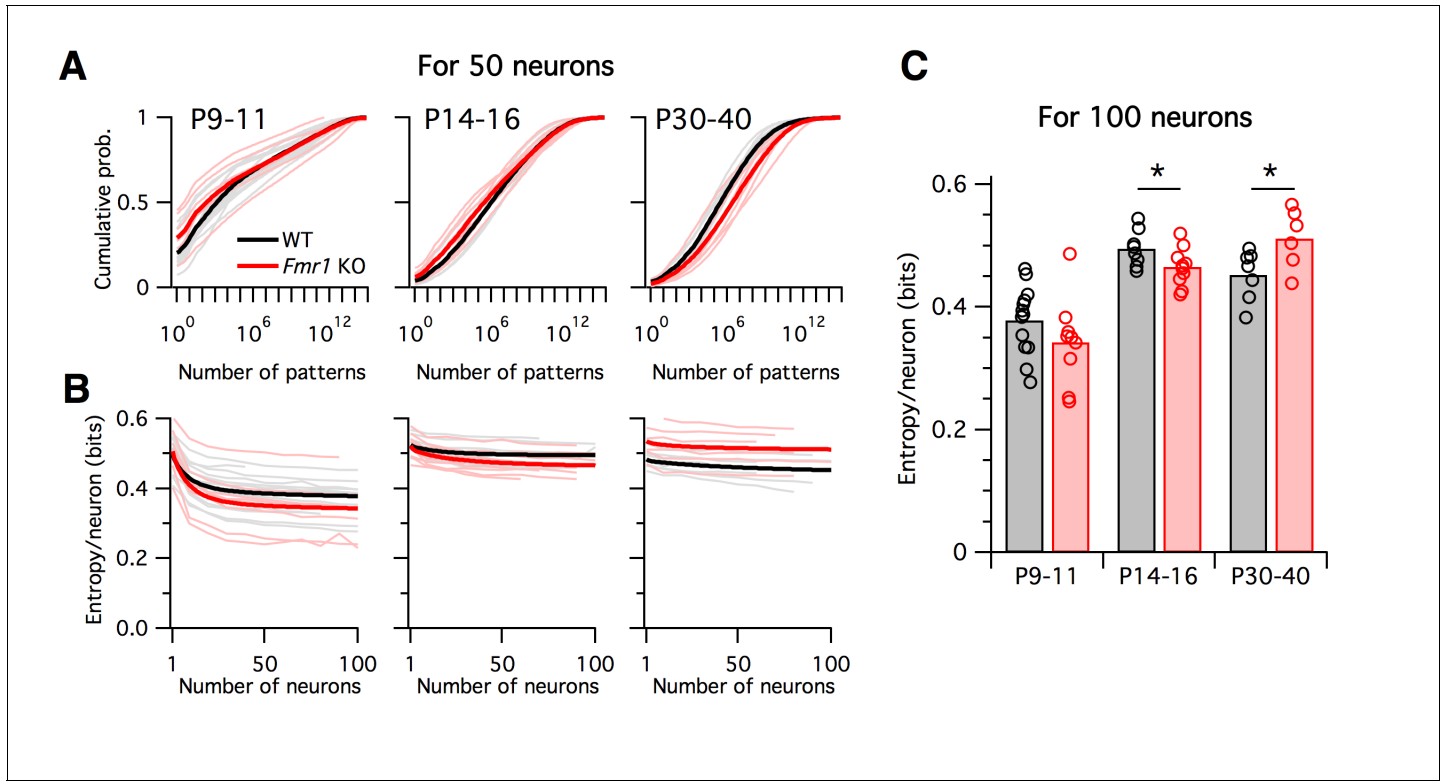

**Figure 7.** Differing trajectories of WT and KO entropy across development. (**A**) Cumulative probability mass as a function of the number of patterns. Patterns ordered from most probable to least probable. Thin lines are mean across many randomly-chosen 50-neuron subsets from a given animal, and thick lines represent means across all animals of a given genotype. (**B**) Entropy per neuron as a function of the number of neurons analyzed. Thin lines are mean across many randomly chosen subsets for a given animal, thick lines are group mean of double exponential fits to the data (see Materials and methods). Age groups (left to right) are as in panel A. (**C**) Estimated entropy/neuron for 100 neuron populations. Circles represent individual animals, bars are group means.
DOI: https://doi.org/10.7554/eLife.26724.011

The following figure supplement is available for figure 7:

**Figure 7–figure supplement 1.** Agreement between entropy estimated from raw data with entropy estimated from samples from fitted logistic models.
DOI: https://doi.org/10.7554/eLife.26724.012

## Discussion

The one-dimensional E/I imbalance model has been widely used for interpreting neural circuit changes observed in animal models of diverse brain disorders (*Bateup et al., 2011*; *Dani et al., 2005*; *Gibson et al., 2008*; *Kehrer et al., 2008*; *Wallace et al., 2012*). In the case of Fragile-X syndrome, the hyperexcitability prediction of the E/I imbalance model is consistent with many of the symptoms of the disease (e.g. seizures, hyperarousal, hyperactivity, hypersensitivity to sensory stimuli) and the known pathogenic defects implicated in *Fmr1* KO mice (diminished GABA signaling, exaggerated intrinsic excitability, increased neuronal firing rates; reviewed by *Contractor et al., 2015*). Here, we tested the hypothesis that the E/I imbalance model can account for alterations in other neural activity statistics beyond the mean firing rates; however, our results demonstrated that it was inadequate. The model was too inflexible to account for the joint alterations in both neural firing rates and correlations observed in Fragile-X model mice. This suggests that future studies of brain disorders may need to consider higher-dimensional models of neural circuit dysfunction.

To test how cellular components affect their circuit function, we built computational models of mouse L2/3 somatosensory cortex at two levels of abstraction: a detailed, 100-parameter biophysical model, and a two-parameter logistic response model. The purpose of the detailed model was to build a representation of the circuit where each parameter has a one-to-one mapping with something that could be experimentally measured in a real animal - indeed many of these parameters have been shown to be altered in *Fmr1* KO mice. The purpose of the logistic model was different: it simple enough to be both derivable from the complex model, and provide a direct link with measurable activity variables in our in vivo Ca2 +imaging data, firing rates and correlations. The disadvantages of detailed models are that they contain many parameters, and so are hard to constrain to data – in this case it was only possible because of the large dataset from Petersen et al., for P17-22 WT mice (*Avermann et al., 2012*; *Lefort et al., 2009*; *Tomm et al., 2014*). The disadvantages of the simple 2D model is that its logistic input-output structure implies a very strong and specific assumption about the functional purpose of the circuit – to generate single spikes across a subset of neurons. Although this may be a physiologically relevant computation for this particular brain circuit (*Clancy et al., 2015*; *Kerr et al., 2007*; *Sato et al., 2007*), it is not immediately obvious how to extend this approach to include temporal correlations, for example, or to apply it to other brain circuits where we may have less insight into their natural computations. Nevertheless, our approach demonstrates a new way to tackle such problems.

After building the detailed computational model of L2/3 of mouse somatosensory cortex (*Figure 2*), we asked how sensitive the spiking responses of the overall circuit were to changes in its underlying neural components, many of which are known to be altered in *Fmr1* KO mice (*Bureau et al., 2008*; *Gibson et al., 2008*; *Gonçalves et al., 2013*; *Harlow et al., 2010*; *Hays et al., 2011*; *Paluszkiewicz et al., 2011*; *Patel et al., 2013*; *Testa-Silva et al., 2012*). We found that while changing some neural parameters did have a large effect, changing other parameters had little or no effect on circuit function (*Figure 3B*). This redundancy property has been reported as widely prevalent in computational models of biological systems (*Gutenkunst et al., 2007*; *O'Leary et al., 2015*). Its existence has two important implications for studies of brain disorders: first, many of the physiological component changes discovered in animal models may be entirely benign at the circuit level. Second, any treatment designed to correct circuit function is free to push the system by arbitrary amounts along insensitive directions in parameter space without consequence, as long as it makes the correct perturbations along the sensitive directions. The insensitive directions form a null space, which is a subspace of the parameter space.

An important caveat to our parameter sensitivity analysis is that it was linear and local to a particular point in the high-dimensional model parameter space, corresponding to WT P17-22 mice. Since the circuit dynamics are nonlinear, it is likely that the particular parameter sensitivities would be different in other parts of parameter space, especially near bifurcations where qualitatively different dynamics emerge (*Hirsch et al., 2013*). However, as long as the redundancy property is widely preserved, as suggested by studies on computational models of other biological systems (*Fisher et al., 2013*; *Gutenkunst et al., 2007*; *Machta et al., 2013*; *Panas et al., 2015*), then our conclusions for brain disorders remain valid.

In addition to the varying magnitudes of circuit components' effect on circuit function, we also found that different components shifted the circuit input-output function in different directions, as

defined by our 2D logistic response model (*Figures 1* and *2*). Even circuit parameters that are nominally of the same type, such as the strength of glutamatergic synapses between excitatory (E) neurons in L4 to E neurons in L2/3 or synaptic strength between E neurons within L2/3, had qualitatively different effects on the circuit response to stimulation (*Figure 3*). According to the standard E/I imbalance model (*Rubenstein and Merzenich, 2003*), both of these parameters should have similar effects on circuit function; but according to the logistic response model we studied, their differing effects on slope and threshold parameters must necessarily lead to different magnitudes of change in neural firing rates and correlations (*Figure 4C*). Indeed, no 1-dimensional model of circuit function could ever capture the heterogeneity in parameter sensitivities that we observed (*Figure 3B*).

Next, we fit the parameters of the logistic response model to match the in vivo firing statistics of neural populations from WT and *Fmr1* KO mice of varying age (*Figure 5*). Previous studies had found that neural correlations decrease during development (*Golshani et al., 2009*; *Rochefort et al., 2009*), and that early postnatal *Fmr1* KO mice had higher correlations and firing rates than WT mice (*Gonçalves et al., 2013*; *La Fata et al., 2014*). Circuit hypersynchrony may be a general defect in autism disorders, as it is also found in mouse models of Rett syndrome (*Lu et al., 2016*). However, the relationship between these changes in firing statistics and the underlying neural circuit components were unclear. Our logistic model helps bridge this gap, leading to two findings: first, the direction of circuit parameter change from WT to KO was opposite in young (P9–11 and P14–16) versus mature (P30–40) animals (*Figure 5C*). Similar opposing switches in sensory cortex properties with age were also recently reported in *Fmr1* KO and WT rats (*Berzhanskaya et al., 2016*). Second, we found that the sensitivity of neural firing rates and correlations to changes in underlying circuit components depends on developmental age (*Figure 6*). Taken together, these findings imply that qualitatively different interventions may be needed at different stages of development in Fragile-X, and perhaps other neurodevelopmental disorders, to shift cortical circuit function towards typical wild-type operation.

Spontaneous, intrinsic activity is ubiquitously present in mammalian cerebral cortex. It is highly structured at multiple spatiotemporal scales (*Mitra et al., 2015*; *Ringach, 2009*) and interacts strongly with the signals evoked by sensory stimulation (*Ringach, 2009*). Cellular-resolution recordings in animals have shown that the patterns of spontaneous activity in neural populations are representative of the ensemble of activity patterns used by the brain to represent sensory stimuli (*Berkes et al., 2011*; *Luczak et al., 2009*; *Miller et al., 2014*). Here, we found that the entropy of spontaneous activity in WT mouse somatosensory cortex follows an inverted-U shaped trajectory across development, and that this trajectory is dramatically altered in the *Fmr1* KO mouse model of Fragile-X (*Figure 7*). Although we saw no reliable differences across genotypes in early postnatal animals (P9–11), *Fmr1* KO animals showed lower entropy than WT after the second postnatal week (P14–16), while surprisingly switching to show higher entropy than WT in adult (P30–40). Notably, this switch in the direction of entropy change from WT to KO during development mirrors the reversing we saw in logistic model parameter changes in *Figure 5C*. Together, these findings suggest a perturbed trajectory of cortical development during the critical period in *Fmr1* KO mice (*Meredith et al., 2012*). However, our results cannot distinguish whether the observed perturbation in L2/3 activity statistics reflects a developmental delay, or a permanently altered developmental trajectory. Further studies at later developmental time points are needed.

What is the functional significance of these shifts in population entropy? Previous work suggested that the entropy of neural circuit activity may be optimally tuned at intermediate levels as a trade-off between maximizing representational capacity at high entropy, versus maintaining error correction and regularization at low entropy (*Schneidman et al., 2006*). These properties can also be thought of as trading off between discrimination and generalization, respectively (*Qian and Lipkin, 2011*). If we assume that WT mice are optimally tuned, our findings predict that young *Fmr1* KO mice should show poorer somatosensory discrimination in behavioral tasks than wild-type animals, while in contrast adult *Fmr1* KO mice should perform more poorly on tasks involving generalization across somatosensory stimuli.

If the unidimensional E/I imbalance model is not sufficiently rich to capture the circuit changes observed in neurodevelopmental disorders, what should replace it? How many dimensions or degrees of freedom should a working model for a brain disorder have? Theoretical neuroscientists have long studied E/I balance in generic models of recurrent neural circuits (*Brunel and Brunel, 2000*; *Tsodyks and Sejnowski, 1995*). These models have uncovered important distinctions

between 'loose' balanced regimes, where E and I inputs to a neuron are equal only on average, and fine-tuned 'tight' balanced regimes where E and I inputs to a neuron track each other closely on fast timescales (*Denève and Machens, 2016*; *Hennequin et al., 2017*). In principle, these generic models could be used to investigate multidimensional E/I imbalances in brain disorders (*Vogels and Abbott, 2007*). However, it is currently difficult to directly fit these many-parameter network models to data (although see *Arakaki et al., 2017*; *Fisher et al., 2013*; *Stringer et al., 2016*), and they are agnostic to circuit function. Instead we suggest an alternative, complementary approach: start by assuming a computational function for the particular neural circuit under study, then work backwards to design a model that is both sophisticated enough to capture the key information processing features of the circuit, but simple enough to interpret and link to physiological data. In this study, we considered a two-parameter model of L2/3 somatosensory cortex's input-output function, which could account for both neural firing rates and correlations. Other brain circuits may demand models with more degrees of freedom. Crucially, the most informative models need not be those that include the highest level of physiological detail. All models are ultimately wrong in the sense that they make abstractions about their underlying parts, and detailed models carry the additional burden of fitting many parameters, which may be difficult to adequately constrain (*O'Leary et al., 2015*). Nonetheless, some models are useful (*Box, 1979*).

One potential use of simple parametric circuit models such as the ones we employed here may be as a tool for rationally designing candidate intervention compounds and then screening their effects on neural population activity. For example, the current study could have been extended to fit the logistic model to neural activity data from another cohort of *Fmr1* KO mice that had received a candidate treatment, then ask if the fitted model parameters were closer in value to those from WT animals or *Fmr1* KO controls. Approaches like this could complement the traditional strategy of designing drugs based on reversing molecular deficits and then assessing the drug's impact on animal model behavior. Indeed, our results suggest that given the multi-dimensionality of circuit properties, it may prove difficult or impossible to find a single compound that can correctly reverse deficits at any age. This scenario might require a combination of drugs chosen to push circuit-level properties towards the 'correct' region of parameter space. The framework we have introduced in this study can facilitate this type of high-dimensional intervention analysis for diverse neurodevelopmental disorders.

## Materials and methods

### Key resources table

| Reagent type (species) or resource | Designation | Source or reference | Identifiers |
|---|---|---|---|
| strain, strain background (mus musculus) | c57bl/6J strain of wild type mice | Jackson Labs | IMSR_JAX:000664 |
| genetic reagent (mus musculus) | Fmr1 knockout mouse on a c57 background | William Greenough (originally from Dutch-Belgian Fragile X Consortium) | RRID:MGI:2665400 |
| chemical compound, drug | OGB1 AM (Oregon Green BAPTA-1 AM) | Molecular Probes (ThermoFisher Scientific) | |
| Software, algorithm | ImageJ | NIH | RRID:SCR_003070 |
| software, algorithm | BRIAN Simulator | http://briansimulator.org | RRID:SCR_002998 |
| software, algorithm | MATLAB | http://www.mathworks.com/products/matlab | RRID:SCR_001622 |

### Mouse in vivo calcium imaging recording

All $Ca^{2+}$ imaging data were published previously (*Gonçalves et al., 2013*). Briefly, data were collected from male and female C57Bl/6 wild-type and *Fmr1* KO mice at P9–40. For each group the animal numbers were: P9-11, n = 13 WT and n = 9 *Fmr1* KO; P14-16, n = 8 WT and n = 10 Fmr1 KO; P30-40, n = 7 WT and n = 6 *Fmr1* KO. There were variations in the number of cells recorded from each animal. The range of cell numbers for each group were: P9-11, 49–198 cells in WT and 84–144 cells in *Fmr1* KO; P14-16, 65–119 cells in WT and 40–149 cells in *Fmr1* KO; P30-40, 60–114 cells in WT and 69–105 cells in *Fmr1* KO. Mice were anesthetized with isoflurane, and a cranial window was

fitted over primary somatosensory cortex by stereotaxic coordinates. Mice were then transferred to a two-photon microscope and headfixed to the stage while still under isoflurane anesthesia. 2–4 injections of the $Ca^{2+}$ sensitive Oregon-Green BAPTA-1 (OGB) dye and sulforhodamine-101 (to visualize astrocytes) were injected 200 um below the dura. Calcium imaging was performed using a Ti-Sapphire Chameleon Ultra II laser (Coherent) tuned to 800 nm. Imaging in unanesthetized mice began within 30–60 min of stopping the flow of isoflurane after the last OGB injection. Images were acquired using ScanImage software (*Pologruto et al., 2004*) written in MATLAB (MathWorks; RRID: SCR_001622). Whole-field images were collected using a 20 × 0.95 NA objective (Olympus) at an acquisition speed of 3.9 Hz (512 × 128 pixels). Several 3 min movies were concatenated and brief segments of motion artifacts were removed (always <10 s total). Data were corrected for x–y drift. Cell contours were automatically detected and the average ΔF/F signal of each cell body was calculated at each time point. The baseline F for each cell was calculated as the mean ROI fluorescence across the entire 3 min timeseries. Neuronal and neuropil signals were analyzed separately and astrocytic signals were excluded from analysis. Each ΔF/F trace was low-pass filtered using a Butterworth filter (coefficient of 0.16) and deconvolved with a 2 s single-exponential kernel (*Yaksi and Friedrich, 2006*). To remove baseline noise, the standard deviation of all points below zero in each deconvolved trace was calculated, multiplied by two, and set as the positive threshold level below which all points in the deconvolved trace were set to zero. Estimated firing rates of the neurons, $r_i(t)$, were then obtained by multiplying the deconvolved trace by a factor of 78.4, previously derived empirically from cell-attached recordings in vivo (*Golshani et al., 2009*).

## Computational methods

Data analysis and logistic model calculations were done using MATLAB (Mathworks; RRID:SCR_001622). All simulation codes are available online at https://github.com/cianodonnell/ODonnelletal_2017_imbalances (copy archived at https://github.com/elifesciences-publications/ODonnelletal_2017_imbalances), and the population tracking model code (*O'Donnell et al., 2017*) is available at https://github.com/cianodonnell/PopulationTracking. (copy archived at https://github.com/elifesciences-publications/PopulationTracking)

## Detailed layer 2/3 model simulations

Layer 2/3 model simulations (*Figures 1* and *2*) were implemented with the Python-based simulator Brian 2 (http://briansimulator.org; RRID:SCR_002998) (*Goodman and Brette, 2009*), and results analyzed with MATLAB (Mathworks; RRID:SCR_001622). The model consisted of four populations of reciprocally connected leaky integrate-and-fire neurons representing a L2/3 somatosensory barrel circuit: 1700 excitatory neurons, 70 PV inhibitory neurons, 115 $5HT_{3A}R$ inhibitory neurons, and 45 SOM inhibitory neurons, driven by a separate population of 1500 excitatory spike sources representing input from L4. Cell numbers were estimated by combining layer-specific excitatory and inhibitory cell count information from (*Lefort et al., 2009*) with the approximate percentages of the three inhibitory cell groups given by (*Petersen and Crochet, 2013*). The voltage $V$ of each neuron evolved as

$$\frac{dV}{dt} = \left( R_{in} \left( g_e \left( E_{rev,e} - V \right) + g_i \left( E_{rev,i} - V \right) \right) - \left( V - V_{rest} \right) \right) / \tau_m$$

where $R_{in}$ is the input resistance, $E_{rev,e}$ and $E_{rev,i}$ are the excitatory and inhibitory synaptic reversal potentials respectively, $\tau_m$ is the membrane time constant, and $g_e$ and $g_i$ are the summed excitatory and inhibitory synaptic input conductances, respectively. Between input events the total excitatory synaptic conductance $g_e$ evolved in time according to the equation

$$\frac{dg_e}{dt} = -g_e / \tau_{syn,e}$$

where $\tau_{syn,e}$ is the excitatory synaptic time constant. Similar equations governed the inhibitory conductances. When a spike arrived at a synapse, a Bernoulli random number was drawn with release probability set according to the particular synaptic connection type. If this number was equal to one, then the total synaptic conductance for that neuron was instantaneously incremented by the specific amplitude of the chosen conductance for that individual synapse, indexed $j$: $g_e \rightarrow g_e + \bar{g}_j$.

All synaptic connections were formed probabilistically by drawing independent random Bernoulli variables with connection type-specific probabilities. Synaptic PSP amplitudes were drawn independently for each synapse from a log-normal distribution constrained by the experimentally reported mean and median values for each particular connection type. The maximum post-synaptic potential amplitude was set to 8 mV. Synapses in the model were conductance-based, but since synaptic strengths reported in the literature were typically in terms of EPSP/IPSP amplitude, in accordance with how the experiments were performed (*Avermann et al., 2012*), we set each maximal synaptic conductance as the value needed to generate a PSP of the desired amplitude when the target neuron started at resting potential in the case of EPSPs or $-55$ mV in the case of IPSPs, which we computed analytically. Refractory periods were calculated as the inter-spike-interval corresponding to the maximal experimentally reported firing rate. Release probability and synaptic strength values for unconnected neurons are excluded from *Table 1*. Excitatory synaptic time constants were set at 2 ms, which is typical for the fast component of AMPA receptor responses, but could not be estimated from the PSP statistics in (*Avermann et al., 2012*) because of masking by the slower membrane time constant. The mathematical form of our model meant that inhibitory synaptic time constants needed to be equal for all incoming inhibitory synapses to a neuron. We set these to 40 ms for E, $5HT_{3A}R$ and SOM neurons and 16 ms for PV neurons, which were the typical values of the IPSP decay time constants in the (*Avermann et al., 2012*) dataset. Due to lack of direct data for this circuit, connection probabilities for synapses from L4 E neurons to E, PV and SOM L2/3 neurons was set to a reasonable cortical value of 0.15, while $5HT_{3A}R$ neurons did not receive any input from L4 (*Gentet et al., 2012*). Similarly due to a lack of direct data, we set synaptic release probabilities for connections from L4 to L2/3 neurons to a typical cortical value of 0.25, while mean and median L4 excitatory PSP amplitudes onto L2/3 PV and SOM were set to 0.8 and 0.48 mV, respectively, to match reported data for L4 EPSP amplitudes onto L2/3 E neurons (*Lefort et al., 2009*). The differential equations were solved using the forward Euler method with an integration timestep of 0.01 ms. Each simulation run was 50 ms long, during which we recorded whether or not each neuron responded. In the rare cases where a neuron spiked more than once, we disregarded the extra spikes. L4 neuron dynamics were not explicitly simulated, but instead modeled only as a set of output spike trains. After selecting the subset of active L4 neurons, spike times were drawn randomly from a Gaussian distribution with standard deviation of 2 ms. We repeated the simulations 10 times for this identical input pattern to average over the noise due to probabilistic vesicle release. We repeated this procedure further 10 times for different random allocations of the 'ON' inputs. Then, a neuron's ON probability was defined as the fraction of these $10 \times 10 = 100$ simulations for which it responded with one or more spikes. Finally, we repeated the entire procedure for varying levels of L4 input sparsity.

For the simulations presented in *Figure 3* we varied only 76 model parameters, which is 24 less than the total number of 100 model parameters listed in *Table 1*. We excluded the four neuronal refractory periods (because in almost all simulations each neuron spiked a maximum of once, making the refractory period irrelevant), and the six connection probabilities that were fixed at zero. Finally, we grouped together the mean and median PSP amplitudes for each of the fourteen non-zero synaptic connections, so that both parameters were increased or decreased by the same fraction in tandem. Together these choices reduced the number of test parameters from 100 to 76.

For all parameters that naturally range from 0 upwards, such as the number of neurons or release probability, we increased or decreased their values during testing in the most intuitive way, by adding ±20% of the baseline value. However, this method was less useful for other parameters, such as cell resting voltage, for which we reasoned it made more sense to scale relative to another parameter, such as spike threshold. As a result, we varied (1) resting voltage relative to its difference from spike threshold; (2) spike threshold relative to its difference with resting voltage; (3) excitatory synaptic reversal potentials relative to resting voltage; (4) inhibitory synaptic reversal potentials relative to spike threshold.

Source [1] is (*Lefort et al., 2009*), [2] is (*Avermann et al., 2012*), [3] is (*Fanselow et al., 2008*), [4] is (*Kinnischtzke et al., 2012*), [5] is (*Fino and Yuste, 2011*). $N$ is number of neurons, $V_{rest}$ is resting potential, $V_{th}$ is spike voltage threshold, $R_{in}$ is input resistance, $t_{ref}$ is refractory period, $\tau_m$ is the membrane time constant, $\tau_{syn}$ is the synaptic time constant with the first subscript indicating the postsynaptic neuron type and the second subscript the neurotransmitter type of the presynaptic neuron ($e$ or $i$), $E_{rev}$ is the synaptic reversal potential, $p_{con}$ is the synaptic connection probability,

**Table 1.** L2/3 computational circuit model parameters and mean slope and threshold shifts.

| Parameter | Value | Source | +20% effect on slope, thresh | Parameter | Value | Source | +20% effect on slope, thresh |
|---|---|---|---|---|---|---|---|
| $N_E$ | 1700 | [1] | slope: $6.4 \times 10^{-3}$<br>thresh: 5.16 | $pcon_{I5htE}$ | 0.465 | [2] | slope: $-3.25 \times 10^{-3}$<br>thresh: −0.103 |
| $N_{Ipv}$ | 70 | [1] | slope: $4.3 \times 10^{-3}$<br>thresh: −1.51 | $pcon_{I5htIpv}$ | 0.38 | [2] | slope: $-7.13 \times 10^{-3}$<br>thresh: −0.298 |
| $N_{I5ht}$ | 115 | [1] | slope: −0.0104<br>thresh: −0.58 | $pcon_{I5htI5ht}$ | 0.38 | [2] | slope: $6.07 \times 10^{-4}$<br>thresh: 0.175 |
| $N_{Isom}$ | 45 | [1] | slope: −0.001<br>thresh: −0.83 | $pcon_{I5htIsom}$ | 0 | No data | Not tested |
| $N_{EL4}$ | 1500 | [1] | slope: −0.106<br>thresh: 3.43 | $pcon_{IsomE}$ | 0.5 | [5] | slope: $-6.29 \times 10^{-3}$<br>thresh: −0.8919 |
| $Vrest_E$ | −68 mV | [2] | slope: 0.049<br>thresh: −6.09 | $pcon_{IsomIpv}$ | 0 | No data | Not tested |
| $Vrest_{Ipv}$ | −68 mV | [2] | slope: −0.016<br>thresh: −0.963 | $pcon_{IsomI5ht}$ | 0 | No data | Not tested |
| $Vrest_{I5ht}$ | −62 mV | [2] | slope: $-9.48 \times 10^{-4}$<br>thresh: 0.205 | $pcon_{IsomIsom}$ | 0 | No data | Not tested |
| $Vrest_{Isom}$ | −57 mV | [3] | slope: $3.58 \times 10^{-3}$<br>thresh: 0.034 | $prel_{EL4E}$ | 0.25 | No data | slope: −0.112<br>thresh: 4.21 |
| $Vth_E$ | −38 mV | [2] | slope: $-3.76 \times 10^{-3}$<br>thresh: −0.381 | $prel_{EL4Ipv}$ | 0.25 | No data | slope: 0.0103<br>thresh: 0.682 |
| $Vth_{Ipv}$ | −37.4 mV | [2] | slope: $3.87 \times 10^{-3}$<br>thresh: 0.221 | $prel_{EL4Isom}$ | 0.25 | No data | slope: $-2.06 \times 10^{-3}$<br>thresh: −0.537 |
| $Vth_{I5ht}$ | −36 mV | [2] | slope: $-1.12 \times 10^{-3}$<br>thresh: 0.013 | $prel_{EE}$ | 0.25 | No data | slope: $4.99 \times 10^{-3}$<br>thresh: 6.074 |
| $Vth_{Isom}$ | −40 mV | [3] | slope: $-3.26 \times 10^{-3}$<br>thresh: −0.083 | $prel_{EIpv}$ | 0.25 | No data | slope: $-4.04 \times 10^{-4}$<br>thresh: 0.163 |
| $Rin_E$ | 160 MΩ | [2] | slope: $1.95 \times 10^{-3}$<br>thresh: −0.283 | $prel_{EI5ht}$ | 0.25 | No data | slope: $-9.32 \times 10^{-3}$<br>thresh: −0.532 |
| $Rin_{Ipv}$ | 100 MΩ | [2] | slope: $-8.38 \times 10^{-3}$<br>thresh: −0.283 | $prel_{EIsom}$ | 0.25 | No data | slope: $-7.32 \times 10^{-3}$<br>thresh: −0.3847 |
| $Rin_{I5ht}$ | 200 MΩ | [2] | slope: $-3.87 \times 10^{-3}$<br>thresh: −0.653 | $prel_{IpvE}$ | 0.25 | No data | slope: $1.03 \times 10^{-2}$<br>thresh: −0.941 |
| $Rin_{Isom}$ | 250 MΩ | [4] | slope: $7.55 \times 10^{-3}$<br>thresh: 0.465 | $prel_{IpvIpv}$ | 0.25 | No data | slope: $1.21 \times 10^{-3}$<br>thresh: −0.061 |
| $\tau m_E$ | 28 ms | [2] | slope: $-9.59 \times 10^{-3}$<br>thresh: 0.268 | $prel_{IpvI5ht}$ | 0.25 | No data | slope: $-4.73 \times 10^{-3}$<br>thresh: 0.011 |
| $\tau m_{Ipv}$ | 21 ms | [2] | slope: $4.82 \times 10^{-3}$<br>thresh: 0.027 | $prel_{I5htE}$ | 0.25 | No data | slope: $-1.63 \times 10^{-3}$<br>thresh: −0.379 |
| $\tau m_{I5ht}$ | 10 ms | [2] | slope: $-5.09 \times 10^{-3}$<br>thresh: −0.302 | $prel_{I5htIpv}$ | 0.25 | No data | slope: $-3.41 \times 10^{-3}$<br>thresh: −0.262 |
| $\tau m_{Isom}$ | 30 ms | [4] | slope: $-2.89 \times 10^{-3}$<br>thresh: −0.159 | $prel_{I5htI5ht}$ | 0.25 | No data | slope: $3.05 \times 10^{-3}$<br>thresh: 0.123 |
| $tref_E$ | 55.5 ms | [2] | Not tested | $p_{rel,IsomE}$ | 0.25 | No data | slope: $-2.13 \times 10^{-4}$<br>thresh: −0.65 |
| $tref_{Ipv}$ | 5.4 ms | [2] | Not tested | $w_{EL4E,mean}$ | 0.8 mV | [1] | slope: −0.142<br>thresh: 0.342 |
| $tref_{I5ht}$ | 21.3 ms | [2] | Not tested | $w_{EL4E,median}$ | 0.48 mV | [1] | slope: −0.142<br>thresh: 0.342 |
| $tref_{Isom}$ | 20 ms | [3] | Not tested | $w_{EL4Ipv,mean}$ | 0.8 mV | $=wEL_{EL4E}$ | slope: $3.61 \times 10^{-3}$<br>thresh: $6.54 \times 10^{-3}$ |
| $\tau syn_{E,e}$ | 2 ms | Typical | slope: $1.37 \times 10^{-2}$<br>thresh: −1.79 | $w_{EL4Ipv,median}$ | 0.48 mV | $=wEL_{EL4E}$ | slope: $3.61 \times 10^{-3}$<br>thresh: $6.54 \times 10^{-3}$ |
| $\tau syn_{E,i}$ | 40 ms | [2] | slope: $-7.29 \times 10^{-3}$<br>thresh: 0.48 | $w_{EL4Isom,mean}$ | 0.8 mV | $=wEL_{EL4E}$ | slope: $5.05 \times 10^{-3}$<br>thresh: −0.329 |

*Table 1 continued on next page*

*Table 1 continued*

| Parameter | Value | Source | +20% effect on slope, thresh | Parameter | Value | Source | +20% effect on slope, thresh |
|---|---|---|---|---|---|---|---|
| $\tau syn_{Ipv,e}$ | 2 ms | Typical | slope: $-9.79 \times 10^{-3}$<br>thresh: $-0.477$ | $w_{EL4Isom,median}$ | 0.48 mV | $=wEL4E$ | slope: $5.05 \times 10^{-3}$<br>thresh: $-0.329$ |
| $\tau syn_{Ipv,i}$ | 16 ms | [2] | slope: $1.56 \times 10^{-3}$<br>thresh: $-0.097$ | $w_{EE,mean}$ | 0.37 mV | [2] | slope: $7.44 \times 10^{-3}$<br>thresh: $5.34$ |
| $\tau syn_{I5ht,e}$ | 2 ms | Typical | slope: $4.52 \times 10^{-3}$<br>thresh: $-0.047$ | $w_{EE,median}$ | 0.2 mV | [2] | slope: $7.44 \times 10^{-3}$<br>thresh: $5.34$ |
| $\tau syn_{I5ht,i}$ | 40 ms | [2] | slope: $-3.82 \times 10^{-3}$<br>thresh: $-0.387$ | $w_{EIpv,mean}$ | 0.82 mV | [2] | slope: $-3.77 \times 10^{-4}$<br>thresh: $-0.297$ |
| $\tau syn_{Isom,e}$ | 2 ms | Typical | slope: $-0.0126$<br>thresh: $-8.82 \times 10^{-3}$ | $w_{EIpv,median}$ | 0.68 mV | [2] | slope: $-3.77 \times 10^{-4}$<br>thresh: $-0.297$ |
| $\tau syn_{Isom,i}$ | 40 ms | [2] | slope: $-4.88 \times 10^{-3}$<br>thresh: $-0.301$ | $w_{EI5ht,mean}$ | 0.39 mV | [2] | slope: $-6.81 \times 10^{-3}$<br>thresh: $-0.46$ |
| $Erev_e$ | 0 mV | Typical | slope: $-0.056$<br>thresh: $1.53$ | $w_{EI5ht,median}$ | 0.19 mV | [2] | slope: $-6.81 \times 10^{-3}$<br>thresh: $-0.46$ |
| $Erev_{Ei}$ | $-68$ mV | $=Vrest_E$ | slope: $3.2 \times 10^{-3}$<br>thresh: $-3.77$ | $w_{EIsom,mean}$ | 0.5 mV | No data | slope: $-3.61 \times 10^{-3}$<br>thresh: $-0.359$ |
| $Erev_{Ipvi}$ | $-68$ mV | $=Vrest_{Ipv}$ | slope: $-0.010$<br>thresh: $-0.617$ | $w_{EIsom,median}$ | 0.4 mV | No data | slope: $-3.61 \times 10^{-3}$<br>thresh: $-0.359$ |
| $Erev_{I5hti}$ | $-62$ mV | $=VrestI_{5ht}$ | slope: $3.8 \times 10^{-3}$<br>thresh: $0.132$ | $w_{IpvE,mean}$ | 0.52 mV | [2] | slope: $6.41 \times 10^{-3}$<br>thresh: $-1.47$ |
| $Erev_{Isomi}$ | $-57$ mV | $=Vrest_{Isom}$ | slope: $-4.2 \times 10^{-3}$<br>thresh: $-0.088$ | $w_{IpvE,median}$ | $-0.29$ mV | [2] | slope: $6.41 \times 10^{-3}$<br>thresh: $-1.47$ |
| $pcon_{EL4E}$ | 0.15 | No data | slope: $-0.121$<br>thresh: $2.99$ | $w_{Ipvlpv,mean}$ | $-0.56$ mV | [2] | slope: $-2.52 \times 10^{-3}$<br>thresh: $-0.345$ |
| $pcon_{EL4Ipv}$ | 0.15 | No data | slope: $1.38 \times 10^{-3}$<br>thresh: $0.029$ | $w_{Ipvlpv,median}$ | $-0.44$ mV | [2] | slope: $-2.52 \times 10^{-3}$<br>thresh: $-0.345$ |
| $pcon_{EL4I5ht}$ | 0 | No data | Not tested | $w_{IpvI5ht,mean}$ | $-0.83$ mV | [2] | slope: $-4.24 \times 10^{-3}$<br>thresh: $-0.266$ |
| $pcon_{EL4Isom}$ | 0.15 | No data | slope: $7.62 \times 10^{-3}$<br>thresh: $-0.245$ | $w_{IpvI5ht,median}$ | $-0.6$ mV | [2] | slope: $-4.24 \times 10^{-3}$<br>thresh: $-0.266$ |
| $pcon_{EE}$ | 0.17 | [2] | slope: $5.86 \times 10^{-3}$<br>thresh: $6.084$ | $w_{I5htE,mean}$ | $-0.49$ mV | [2] | slope: $-3.61 \times 10^{-4}$<br>thresh: $-0.018$ |
| $pcon_{EIpv}$ | 0.575 | [2] | slope: $-1.17 \times 10^{-3}$<br>thresh: $-0.099$ | $w_{I5htE,median}$ | $-0.3$ mV | [2] | slope: $-3.61 \times 10^{-4}$<br>thresh: $-0.018$ |
| $pcon_{EI5ht}$ | 0.24 | [2] | slope: $-6.44 \times 10^{-3}$<br>thresh: $-0.541$ | $w_{I5htIpv,mean}$ | $-0.49$ mV | [2] | slope: $-1.77 \times 10^{-3}$<br>thresh: $-0.187$ |
| $pcon_{EIsom}$ | 0.5 | [5] | slope: $-4.37 \times 10^{-3}$<br>thresh: $-0.27$ | $w_{I5htIpv,median}$ | $-0.15$ mV | [2] | slope: $-1.77 \times 10^{-3}$<br>thresh: $-0.187$ |
| $pcon_{IpvE}$ | 0.6 | [2] | slope: $7.16 \times 10^{-3}$<br>thresh: $-1.049$ | $w_{I5htI5ht,mean}$ | $-0.37$ mV | [2] | slope: $-4.12 \times 10^{-3}$<br>thresh: $-0.416$ |
| $pcon_{IpvIpv}$ | 0.55 | [2] | slope: $-2.61 \times 10^{-3}$<br>thresh: $-0.0456$ | $w_{I5htI5ht,median}$ | $-0.23$ mV | [2] | slope: $-4.12 \times 10^{-3}$<br>thresh: $-0.416$ |
| $pcon_{IpvI5ht}$ | 0.24 | [2] | slope: $-2.81 \times 10^{-3}$<br>thresh: $-0.458$ | $w_{IsomE,mean}$ | $-0.5$ mV | No data | slope: $-0.013$<br>thresh: $-0.984$ |
| $pcon_{IpvIsom}$ | 0 | No data | Not tested | $w_{IsomE,median}$ | $-0.4$ mV | No data | slope: $-0.013$<br>thresh: $-0.984$ |

DOI: https://doi.org/10.7554/eLife.26724.013

*prel* is the synaptic release probability, *w* is the mean or median post-synaptic potential amplitude as indicated. For all neuronal parameters, the subscript indicates the neuron type: *E* is L2/3 excitatory neurons, *Ipv* is PV neurons, *I5ht* is $5HT_{3A}R$ neurons, *Isom* is SOM neurons, and *EL4* is L4 excitatory neurons. For synaptic parameters, the first and second subscripts indicate the pre- and post-synaptic neuron types, respectively.

An important caveat is that although this model may be considered detailed by some measures, it also simplifies many aspects of L2/3 circuit. For example, we assumed that all 5HT$_{3A}$R cells were homogeneous, even though they likely separate into different subclasses with type-specific connectivity (*Gentet, 2012*; *Petersen and Crochet, 2013*). Layer 2 and layer 3 may also consist of distinct cell populations (*Petersen and Crochet, 2013*). Not all likely connections were included in the model (*Dalezios et al., 2002*; *Pfeffer et al., 2013*), and connectivity was assumed to be random, even though it is likely non-random (*Tomm et al., 2014*). Although these choices will likely not affect the conclusions of the current study, they may be important to consider for future work that seeks to understand the biological function of the L2/3 somatosensory microcircuit.

## Logistic model

From the L2/3 circuit model simulations, we numerically estimated the probability $q$ that each neuron in the model fires a spike as a function of the fraction of L4 inputs that were active, $f$. We then used the generalized linear model regression tool 'glmfit' in MATLAB to find the best fit of the two logistic model parameters for each neuron:

$$q(f) = \frac{1}{1 + \exp\left(-\beta\left(f - f_{1/2}\right)\right)}$$

where the parameter $\beta$ represents the slope, and the parameter $f_{1/2}$ represents the fraction of active L4 neurons at which the response probability $q = 0.5$. For clarity of presentation, in the main text we converted this $f_{1/2}$ parameter to what we termed the 'threshold', $f_{thresh}$, which we defined as the fraction of L4 neurons needed to reach a specified spike probability, $q_{thresh}$. Throughout the study, we fixed $q_{thresh} = 0.01$. The threshold is related to $q_{thresh}$ via the inverse of the logistic function

$$f_{thresh} = f_{1/2} + \log\left(\frac{q_{thresh}}{1 - q_{thresh}}\right)/\beta$$

We computed firing rates and pairwise correlations from the logistic model (*Figures 4–5*) in the following way. First, we assumed that the fraction of active L4 neurons is described by a normally distributed random variable with zero mean and unit variance:

$$p(f) = \frac{\exp(-f^2/2)}{\sqrt{2\pi}} = N(0,1)$$

We defined the $\beta$ and $f_{1/2}$ parameters relative to the mean and standard deviation of the input distribution. Since $q$ is a monotonically increasing function of $f$, the probability distribution for $q$ is

$$p(q) = p(f(q))\left|\frac{df}{dq}\right|$$

where $f(q)$ is the inverse of the logistic function $q(f)$ and

$\frac{df}{dq} = \frac{\left(\exp\left(-\beta\left(f - f_{1/2}\right)\right) + 1\right)^2}{\beta\exp\left(-\beta\left(f - f_{1/2}\right)\right)}$. We calculate a neuron's mean firing rate $\mu$ as the expectation of $q$,

$$\mu = \mathbb{E}[q] = \int_0^1 [q \times p(q)]dq = \int_0^1 \left[q \times p(f(q))\left|\frac{df}{dq}\right|\right]dq$$

We calculate the pairwise covariance of two homogeneous neurons driven by a common input $f$ as

$\mathrm{cov} = \mathbb{E}[q^2] - (\mathbb{E}[q])^2 = \mathbb{E}[q^2] - \mu^2 = \int_0^1 \left[q^2 \times p(f(q))\left|\frac{df}{dq}\right|\right]dq - \mu^2$, then find the pairwise correlation by normalizing the covariance by the neurons' shared variance, $\mathrm{var} = \mu(1 - \mu)$.

For fitting the logistic model to the recorded neural firing rates and correlations (*Figure 5*), we considered a population model where the joint probability distribution across threshold and slope was specified by a 2D Gaussian, which has five parameters: threshold mean and s.d., slope mean and s.d., and slope-threshold correlation. The three constraint statistics we considered from the neural population data were the mean neural ON probability, the s.d. of neural ON probabilities, and the mean pairwise correlations. We found the best-fit model parameters for each dataset using

stochastic gradient descent (code available at https://github.com/cianodonnell/ODonnelletal_2017_imbalances). Briefly, the fitting procedure followed: (1) initialize the parameters at a starting guess points, (2) compute the predicted three output firing statistics via numerical integration over the model's probability distributions, (3) compute the fitting error as the summed squared difference between the model output predictions and the target values, (4) generate a new set of parameter values by adding a small perturbation of a zero-mean Gaussian random number to each parameter, (5) compute the new output statistics, (5) recompute the fitting error, (6) if the new error is smaller than the old error, accept the updated parameter values, otherwise reject them and revert to the old parameters, (7) return to step 4 unless the error is lower than the desired tolerance. We checked for fit convergence by sampling a large number of logistic model parameters from the fitted 2D Gaussian, drawing binary samples from these logistic model neurons and computing the ON probability mean and s.d., and mean pairwise correlation from the synthetic binary samples, and comparing the computed statistical values to the original data statistics (*Figure 5 – figure supplement 1*). For the sensitivity analysis presented in *Figure 6*, we numerically computed the partial derivative in mean firing rate and pairwise correlation with respect to the mean slope and mean threshold parameters in the population logistical model, using standard finite difference methods.

## Statistical tests

To avoid parametric assumptions, all statistical tests were done using standard bootstrapping methods with custom-written MATLAB scripts. For example, when assessing the observed difference between two group means $\Delta\mu_{obs}$ we performed the following procedure to calculate a p-value. First, we pool the data points from the two groups to create a null set $S_{null}$. We then construct two hypothetical groups of samples $S_1$ and $S_2$ from this by randomly drawing $n_1$ and $n_2$ samples with replacement from $S_{null}$, where $n_1$ and $n_2$ are the number of data points in the original groups 1 and 2 respectively. We take the mean of both hypothetical sets $\Delta\mu_1$ and $\Delta\mu_2$ and calculate their difference $\Delta\mu_{null} = \Delta\mu_1 - \Delta\mu_2$. We then repeat the entire procedure $10^7$ times to build up a histogram of $\Delta\mu_{null}$. This distribution is always centered at zero. After normalizing, this can be interpreted as the probability distribution $f(\Delta\mu_{null})$ for observing a group mean difference of $\Delta\mu_{null}$ purely by chance if the data were actually sampled from the same null distribution. Then the final p-value for the probability of finding a group difference of at least $\Delta\mu_{obs}$ in either direction is given by $p = \int_{-\infty}^{-\Delta\mu_{obs}} f(\Delta\mu_{null})d\Delta\mu_{null} + \int_{\Delta\mu_{obs}}^{\infty} f(\Delta\mu_{null})d\Delta\mu_{null}$.

For *Figure 5C*, we estimated two-dimensional 95% confidence ellipses for the shift in mean slope-threshold parameters between *Fmr1* KO and WT by computing the sample error variances and covariance through bootstrapping. Then, the 95% confidence ellipse can be computed using the Chi-squared distribution. We plotted the confidence interval ellipse using the MATLAB function error_ellipse.m, downloaded from https://www.mathworks.com/matlabcentral/fileexchange/4705-error-ellipse.

## Conversion from firing rate to ON/OFF probabilities for Ca²⁺ imaging data

For the Ca²⁺ imaging data, we began with estimated firing rate time series $r_i(t)$ for each neuron $i$ recorded as part of a population of $N$ neurons. For later parts of the analysis we needed to convert these firing rates to binary ON/OFF values. This conversion involves a choice. One option would be to simply threshold the data, but this would throw away information about the magnitude of the firing rate. We instead take a probabilistic approach where rather than deciding definitively whether a given neuron was ON or OFF in a given time bin, we calculate the probability that the neuron was ON or OFF by assuming that neurons fire action potentials according to an inhomogeneous Poisson process with rate $r_i(t)$. The mean number of spikes $\lambda_i(t)$ expected in a time bin of width $\Delta t$ is $\lambda_i(t) = r_i(t)\Delta t$. We choose $\Delta t = 1$ s. Under the Poisson model the actual number of spikes $m$ in a particular time bin is a random variable that follows the Poisson distribution $P(m = k) = \lambda^k \exp(-\lambda)/k!$. We considered a neuron active (ON) if it is firing one or more spikes in a given time bin. Hence, the probability that a neuron is ON is $p_{on}(t) = 1 - P(m=0) = 1-\exp(\lambda)$. This approach has two advantages over thresholding: (1) it preserves some information about the magnitude of firing rates and (2) it acts to regularize the probability distribution for the number of neurons active by essentially smoothing nearby values together.

## Entropy estimation for large numbers of neurons

Entropy was estimated by fitting a statistical model we recently developed, called the population tracking model (*O'Donnell, 2017a*), to the binarized $Ca^{2+}$ imaging data. Briefly, the population tracking model fits two aspects of the data: the probability distribution for the number of neurons synchronously active in the population, and also the conditional firing probability that each individual neuron is active given the population count. Hence, the model captures both some aggregate statistics of the population activity, and some aspects of the heterogeneity across neurons. See *O'Donnell, 2017b*) for complete details and validation of the method. Code for fitting the model to data is available at https://github.com/cianodonnell/PopulationTracking.

The entropy/neuron generally decreased with the number of neurons considered as result of the population correlations (*Figure 7B*), so we needed to control for neural population size when comparing data from different experimental groups. On the one hand, we would like to study as large a number of neurons as possible, because we expect the effects of collective network dynamics to be stronger for large population sizes and this may be the regime where differences between the groups emerge. On the other hand, our recording methods allowed us to sample only typically around ∼100 neurons at a time, and as few as 40 neurons in some animals. Hence, we proceeded by first estimating the entropy/neuron in each animal by calculating the entropy of random subsets of neurons of varying size from 10 to 100 (if possible) in steps of 10. For each population size we sampled a large number of independent subsets, calculated the entropy of each. Finally, for each dataset we fit a double exponential function to the estimated entropy/neuron as a function of the number of neurons: $H/N = A*\exp(-b*N)+C*\exp(-d*N)+e$, and used this fit to estimate $H/N$ for 100 neurons.

## Acknowledgements

We thank Timothy O'Leary, Hannes Saal, and Alex Williams for comments on earlier versions of the manuscript. This study was supported by funding from FRAXA Research Foundation, Howard Hughes Medical Institute, Sloan-Swartz Foundation, the Dana Foundation, Developmental Disabilities Translational Research Program grant #20160969 (The John Merck Fund), SFARI grant 295438 (Simons Foundation) and the NIH (NICHD R01HD054453 and NINDS RC1NS068093).

## Additional information

### Funding

| Funder | Grant reference number | Author |
|---|---|---|
| FRAXA Research Foundation | Postdoctoral fellowship | Cian O'Donnell |
| Howard Hughes Medical Institute | | Cian O'Donnell<br>Terrence J Sejnowski |
| Sloan-Swartz | | Cian O'Donnell<br>Terrence J Sejnowski |
| Dana Foundation | | J Tiago Gonçalves<br>Carlos Portera-Cailliau |
| John Merck Fund | 20160969 | Carlos Portera-Cailliau |
| Simons Foundation | 295438 | Carlos Portera-Cailliau |
| National Institute of Neurological Disorders and Stroke | RC1NS068093 | J Tiago Gonçalves<br>Carlos Portera-Cailliau |
| Eunice Kennedy Shriver National Institute of Child Health and Human Development | R01HD054453 | J Tiago Gonçalves<br>Carlos Portera-Cailliau |

The funders had no role in study design, data collection and interpretation, or the decision to submit the work for publication.

## Author contributions
Cian O'Donnell, Conceptualization, Formal analysis, Funding acquisition, Investigation, Methodology, Writing—original draft, Writing—review and editing; J Tiago Gonçalves, Data curation, Investigation, Methodology, Writing—review and editing; Carlos Portera-Cailliau, Resources, Data curation, Supervision, Funding acquisition, Methodology, Writing—review and editing; Terrence J Sejnowski, Conceptualization, Supervision, Funding acquisition, Writing—review and editing

## Author ORCIDs
Cian O'Donnell http://orcid.org/0000-0003-2031-9177
Terrence J Sejnowski https://orcid.org/0000-0002-0622-7391

## Ethics
Animal experimentation: All experiments were conducted according the US National Institutes of Health guidelines for animal research, under an animal protocol (ARC#2007-035) approved by the Chancellor's Animal Research Committee and the Office for the Protection of Research Subjects at the University of California, Los Angeles.

## Decision letter and Author response
Decision letter https://doi.org/10.7554/eLife.26724.017
Author response https://doi.org/10.7554/eLife.26724.018

## Additional files

### Supplementary files
• Transparent reporting form
DOI: https://doi.org/10.7554/eLife.26724.014

### Major datasets
The following previously published dataset was used:

| Author(s) | Year | Dataset title | Dataset URL | Database, license, and accessibility information |
|---|---|---|---|---|
| Gonçalves JT, O'Donnell C, Sejnowski TJ, Portera-Cailliau C | 2017 | Two photon Ca2+ imaging recordings of spontaneous activity from mouse somatosensory cortex in wild-type and Fmr1 knock-out mice from three developmental age groups | http://dx.doi.org/10.6080/K0X63K3X | Available at CRCNS.org |

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
