## [Decision Letter]

Thank you for submitting your article "Multidimensional imbalances in cortical circuit activity in Fragile-X Syndrome mice" for consideration by *eLife*. Your article has been reviewed by four peer reviewers, one of whom, Frances Skinner, is a member of our Board of Reviewing Editors and the evaluation has been overseen by Huda Zoghbi as the Senior Editor. The following individuals involved in review of your submission have agreed to reveal their identity: Bill Lytton (Reviewer #2); Mark D Humphries (Reviewer #4).

The reviewers have discussed the reviews with one another and the Reviewing Editor has drafted this decision to help you prepare a revised submission.

Summary:

This study combines computational modeling and analysis of two-photon Ca^2+^ imaging data from in vivo Fragile-X model mice to test the excitatory-inhibitory (E-I) imbalance theory of circuit dysfunction. The microcircuit model highlights the complex non-linear effects of perturbations to cellular components on circuit function, and combined with their logistic response model, indicates that E-I imbalance theory is not sufficient to explain dissociated effects of firing rate and correlations. Analysis of data from *Fmr1* knock-out mice demonstrated differences in input-output circuit parameters across development, which could not be captured by examining firing rate and pairwise correlations between neurons alone.

Overall, the model and analysis are able to show relationships between firing statistics and underlying circuit parameters.

The demonstration of capturing network-wide changes in dynamics by a neuron's or population's P(spike) function is innovative, and opens up a range of possible applications in both modelling and data analysis – both of which are demonstrated here. A nice bonus is the characterization of the developmental trajectory of the WT and *Fmr1* mice. Collectively, the work attains its goal: to evaluate the relevance of E/I balance, and suggest an alternative.

All of the reviewers found the work to be interesting and felt positive in different ways, but all of the reviewers also felt that extensive rewriting and re-organization was required as several points were confusing and/or unclear along with clarifying various issues. There are three essential aspects that the authors need to address as well as the 5 further comments.

Essential revisions:

A) A re-organization and re-write is required so that the motivation and rationale is clear at various junctures. Some reviewers thought it would be better to present the experiment first (as described at the end of the Introduction, "In this study we compare in vivo….") rather than the model first as it is at present. That is, starting with the problem and working forward rather than starting with some solutions and seeming to look for a problem.

For example, presenting the data first would allow it to be used as a motivator for the model, and make it more accessible to a general neuro audience. However, this then would perhaps seem like P(spike) comes out of the blue. If the order is left as is with the model first, then the motivation needs to be more clearly described and the work should perhaps be presented with less emphasis on Fragile-X and instead a focus on the how models can highlight the simplicity of E-I imbalance theories etc. With this latter option, a title change should be considered.

We leave it to the author to decide how they prefer to re-organize and re-write to present their work in a clear, motivating fashion through the various steps, so that the reader can appreciate and grasp what are the main exciting results. That is, there is a need to bring out the logic of certain choices, details of the analyses, and the motivation for the analyses.

B) Caveats and limitations, as well as advantages and disadvantages of their model should be brought forth in the main text for the reader to appreciate up front, and not only given in the Materials and methods section.

For example, explicitly state why and how a simpler model could/should not be considered, and why they do not expect existing model choices/limitations to not necessarily affect the outcome (e.g., "varying the strength of synapses from 5HT_3A_R inhibitory neurons to E neurons.… has little effect" might possibly be troublesome given the model setup).

Even within the Materials and methods section, more rationale for model details should be provided. Model equations need to be provided.

For the above points, additional subsections in the Results section could be helpful.

C) Models are only reliably replicable if downloaded. The model (microcircuit, logistic response model) should be made available via ModelDB, GitHub or other repository.

The following comments from the four reviewers also need to be addressed.

1) Explicit equations for population model and circuit model should be provided. Integration scheme and time constant for easy reproduction needed.

Also, several connectivities are fixed or not examined at all (e.g., 'typical cortical value of 0.15?' – what is that based on if there is no data? etc.).

While robustness is tackled via a +/-20% adjustment and found to be 'sloppy', this is essentially because a dynamical systems understanding does not exist (e.g., there would be more sensitivity near a bifurcation) as the authors would know. While it is clearly not possible to do a dynamical systems analyses (and certainly true in general for high-dimensional systems), I wonder if the authors could provide some understanding of the essence of their network I/O in some way to understand the sloppiness and parameter fixing or not including certain connectivities. While this could be challenging, I think this is important to bring forth in some way since the authors go on to analyse their model as if it is 'good enough' to represent the biological situation, so it seems glossed over.

Further, this forms the basis of the linkage they are making here, and it was not clear to me. That is, the interpretation of model (and its parameters) analyses relative to the calcium imaging analyses that was being compared was obscured and/or unclear.

2) I am somewhat perplexed by the decision to lead with the model and then introduce fragile X data late in the paper. In general, I would be more interested if the paper could present some problems with understanding fragile X pathophysiology, likely by interpreting it initially by using various levels of straw-man simplistic single-factor thinking with a simple or 2-stage (I/E) logistic-function explicit model. From there, the paper could demonstrate how the more sophisticated model can at least permit assessment of features which are inaccessible in a logistic model. Granted that this final L2/3 model also does not directly predict treatments fragile X, it at least points in the direction from which future solutions will come.

3) Issues regarding the construction of the microcircuit model. The authors constrained their PV and 5HT_3A_R model parameters from the experimental work of Avermann et al., 2012. However, this paper identified two GABAergic populations: fast-spiking (shown to be PV-expressing), and non-fast-spiking (NFS; i.e., everything else). Although a large percentage of these NFS cell are likely the heterogeneous 5HT_3A_R population, they would also include SOM-expressing cells. This assumption that NFS are strictly 5HT_3A_R has led to some perhaps misleading choices in model construction and choices for intrinsic and synaptic parameters. For example, ~40% 5HT_3A_R cells are VIP+ in S1 (Lee et al., 2010), and a primary target of VIP+ interneurons are SOM+ (Pfeffer et al., 2013; Dalezios et al., 2002), but the model has no 5HT_3A_R→ SOM connections. Also, it has 5HT_3A_R→5HT_3A_R connections, but not SOM→PV (Pfeffer et al., 2013)?

Overall, these model choices will likely not affect the main results – an overall varied sensitivity to cellular changes. However, it makes the specifics of how distinct populations are affected in Figure 1 and Figure 2 (and the associated text) difficult to interpret from a biological standpoint. Along those lines, Table 1 describes the circuit model variables and could serve as a resource for modelers and experimentalists alike. However, for clarity, readers should be aware of the assumptions made so that they can use with caution.

4) The authors found that the direction of circuit parameters change in young to mature mice is opposite in KO vs WT, which could not be captured by examining neural activity statistics. Given that there is so much redundancy and variability at the cellular level, how does knowing about these circuit properties help us in terms of mechanisms of change?

The logistic response model also fits the threshold s.d., the slope s.d., and the slope-threshold correlation. How were these parameters affected in development/in KO?

5) It is unclear what we learn from the population-level P(spike) model that we could not learn from the single neuron model. Two things it would be good for the authors to address here:

i) The population-level model lacks motivation – I would expect the fourth paragraph of subsection “Firing rates and correlations from the logistic model” to start with something like "In order to characterise the population, we fitted… because…". Why fit a 5-parameter model to the whole population: why not just fit each neuron with a 2-parameter model, and average over their fits? [Just as was done for the model]. Then one would also obtain the population variation in the slope and threshold parameters, and that variation need not conform to a Gaussian model as is enforced by the 5-parameter model. I suspect there is an underlying issue with having enough data per-neuron. Please elaborate.

ii) Having fitted a population model, not much is done with it. It seems plausible that the variation in slope and threshold (quantified by the SD) could change with development, as the underlying parameter ranges contract or expand. So what is the variation over development, and does it differ between WT and *Fmr1* mice?

---

## [Author Response]

Thank you for submitting your article "Multidimensional imbalances in cortical circuit activity in Fragile-X Syndrome mice" for consideration by eLife. Your article has been reviewed by four peer reviewers, one of whom, Frances Skinner, is a member of our Board of Reviewing Editors and the evaluation has been overseen by Huda Zoghbi as the Senior Editor. The following individuals involved in review of your submission have agreed to reveal their identity: Bill Lytton (Reviewer #2); Mark D Humphries (Reviewer #4).The reviewers have discussed the reviews with one another and the Reviewing Editor has drafted this decision to help you prepare a revised submission.Summary:This study combines computational modeling and analysis of two-photon Ca^2+^ imaging data from in vivo Fragile-X model mice to test the excitatory-inhibitory (E-I) imbalance theory of circuit dysfunction. The microcircuit model highlights the complex non-linear effects of perturbations to cellular components on circuit function, and combined with their logistic response model, indicates that E-I imbalance theory is not sufficient to explain dissociated effects of firing rate and correlations. Analysis of data from Fmr1 knock-out mice demonstrated differences in input-output circuit parameters across development, which could not be captured by examining firing rate and pairwise correlations between neurons alone.Overall, the model and analysis are able to show relationships between firing statistics and underlying circuit parameters.The demonstration of capturing network-wide changes in dynamics by a neuron's or population's P(spike) function is innovative, and opens up a range of possible applications in both modelling and data analysis – both of which are demonstrated here. A nice bonus is the characterization of the developmental trajectory of the WT and Fmr1 mice. Collectively, the work attains its goal: to evaluate the relevance of E/I balance, and suggest an alternative.All of the reviewers found the work to be interesting and felt positive in different ways, but all of the reviewers also felt that extensive rewriting and re-organization was required as several points were confusing and/or unclear along with clarifying various issues. There are three essential aspects that the authors need to address as well as the 5 further comments.Essential revisions:A) A re-organization and re-write is required so that the motivation and rationale is clear at various junctures. Some reviewers thought it would be better to present the experiment first (as described at the end of the Introduction, "In this study we compare in vivo….") rather than the model first as it is at present. That is, starting with the problem and working forward rather than starting with some solutions and seeming to look for a problem.For example, presenting the data first would allow it to be used as a motivator for the model, and make it more accessible to a general neuro audience. However, this then would perhaps seem like P(spike) comes out of the blue. If the order is left as is with the model first, then the motivation needs to be more clearly described and the work should perhaps be presented with less emphasis on Fragile-X and instead a focus on the how models can highlight the simplicity of E-I imbalance theories etc. With this latter option, a title change should be considered.We leave it to the author to decide how they prefer to re-organize and re-write to present their work in a clear, motivating fashion through the various steps, so that the reader can appreciate and grasp what are the main exciting results. That is, there is a need to bring out the logic of certain choices, details of the analyses, and the motivation for the analyses.

On reflection we agree with the reviewers – much of the original manuscript’s logic may have proven difficult for a reader to follow. To address this we have now totally reworked the Introduction, inserted a new Figure 1 that includes the motivating Fragile-X data, and added new text to the Results in several places to help clarify the paper’s logic. We believe that the new version is much improved.

B) Caveats and limitations, as well as advantages and disadvantages of their model should be brought forth in the main text for the reader to appreciate up front, and not only given in the Materials and methods section.For example, explicitly state why and how a simpler model could/should not be considered, and why they do not expect existing model choices/limitations to not necessarily affect the outcome (e.g., "varying the strength of synapses from 5HT_3A_R inhibitory neurons to E neurons.… has little effect" might possibly be troublesome given the model setup).

We have added new sentences to the Results to motivate both the biophysical model (“We chose this level of detail for the model in order to relate experimentally measureable biophysical properties of neurons to their putative role in the circuit at large.”) and logistic model (“This sigmoidal-shaped response profile of simulated L2/3 neurons mimics the spiking response of mouse L2/3 pyramidal cells to extracellular L4 stimulation in vitro (Elstrott et al., 2014), while the sparse, noisy and distributed network responses were reminiscent of in vivo activity following whisker stimulation (Clancy et al., 2015; Kerr et al., 2007).”), and a section comparing the logistic model with the E/I imbalance model: “This 2-D logistic model has two benefits over the 1-D E/I imbalance model: first, its extra degree of freedom allows for richer and more flexible fits to data, and second, by describing an input-output mapping for the L2/3 circuit it can capture some aspects of the computation that the circuit performs for the animal. In contrast, the E/I imbalance model is specified purely in terms of circuit components, and so is agnostic to the circuit’s computational function.” And finally, we added a new paragraph to the Discussion on the same issue.

Even within the Materials and methods section, more rationale for model details should be provided. Model equations need to be provided.

We have now added more rationale and all the model equations to the Materials and methods section.

For the above points, additional subsections in the Results section could be helpful.C) Models are only reliably replicable if downloaded. The model (microcircuit, logistic response model) should be made available via ModelDB, GitHub or other repository.We agree that this is an important step. We have now put all the code online at https://github.com/cianodonnell/ODonnelletal_2017_imbalances and have indicated so at the beginning of the Methods section on computational modelling methods.The following comments from the four reviewers also need to be addressed.1) Explicit equations for population model and circuit model should be provided. Integration scheme and time constant for easy reproduction needed.

For the detailed circuit model we used a timestep of 0.01 ms and the forward Euler integration scheme. These details and the model equations are now mentioned in the Methods.

Also, several connectivities are fixed or not examined at all (e.g., 'typical cortical value of 0.15?' – what is that based on if there is no data? etc.).

We are unsure what the reviewer’s point is here – we varied 76 out of the 100 total circuit model parameters, including those for which there was no direct data, such as the synaptic connections with probability of 0.15 that the reviewer highlights. The logic for not varying the remaining 24 parameters was given in the original Materials and methods section: “We excluded the four neuronal refractory periods (because in almost all simulations each neuron spiked a maximum of once, making the refractory period irrelevant), and the six connection probabilities that were fixed at zero. Finally, we grouped together the mean and median PSP amplitudes for each of the fourteen non-zero synaptic connections, so that both parameters were increased or decreased by the same fraction in tandem. Together these choices reduced the number of test parameters from 100 to 76.”

While robustness is tackled via a +/-20% adjustment and found to be 'sloppy', this is essentially because a dynamical systems understanding does not exist (e.g., there would be more sensitivity near a bifurcation) as the authors would know. While it is clearly not possible to do a dynamical systems analyses (and certainly true in general for high-dimensional systems), I wonder if the authors could provide some understanding of the essence of their network I/O in some way to understand the sloppiness and parameter fixing or not including certain connectivities. While this could be challenging, I think this is important to bring forth in some way since the authors go on to analyse their model as if it is 'good enough' to represent the biological situation, so it seems glossed over.

This is a very good point. We have added a new paragraph to the Discussion section to address this issue:

“An important caveat to our parameter sensitivity analysis is that it was linear and local to a particular point in the high-dimensional model parameter space, corresponding to WT P17-22 mice. Since the circuit dynamics are nonlinear, it is likely that the particular parameter sensitivities would be different in other parts of parameter space, especially near bifurcations where qualitatively different dynamics emerge (Hirsch et al., 2013). However, as long as the redundancy property is widely preserved, as suggested by studies on computational models of other biological systems (Fisher et al., 2013; Gutenkunst et al., 2007; Machta et al., 2013; Panas et al., 2015), then our conclusions for brain disorders remain valid.”

Further, this forms the basis of the linkage they are making here, and it was not clear to me. That is, the interpretation of model (and its parameters) analyses relative to the calcium imaging analyses that was being compared was obscured and/or unclear.

This is also a good point. We consider this to be the purpose of the logistic model, which acts as a bridge between the biophysical model and the data. This bridge is necessary because it is not tractable to fit the biophysical model to our calcium imaging data. We have tried hard to clarify this relationship in the new version of the manuscript.

2) I am somewhat perplexed by the decision to lead with the model and then introduce fragile X data late in the paper. In general, I would be more interested if the paper could present some problems with understanding fragile X pathophysiology, likely by interpreting it initially by using various levels of straw-man simplistic single-factor thinking with a simple or 2-stage (I/E) logistic-function explicit model. From there, the paper could demonstrate how the more sophisticated model can at least permit assessment of features which are inaccessible in a logistic model. Granted that this final L2/3 model also does not directly predict treatments fragile X, it at least points in the direction from which future solutions will come.

This reviewer seems to have misinterpreted the point of the paper: our goal was not to “understand fragile-x pathophysiology” directly, but to test whether the E/I imbalance model is flexible enough to account for the neural activity alterations in any brain disorder, taking *Fmr1* KO mice as an example case. We do appreciate that this logic was not clear in the original manuscript and have taken care to emphasize our exact line of thinking in the new version.

3) Issues regarding the construction of the microcircuit model. The authors constrained their PV and 5HT_3A_R model parameters from the experimental work of Avermann et al., 2012. However, this paper identified two GABAergic populations: fast-spiking (shown to be PV-expressing), and non-fast-spiking (NFS; i.e., everything else). Although a large percentage of these NFS cell are likely the heterogeneous 5HT_3A_R population, they would also include SOM-expressing cells.

When we were designing the model we were unclear about these same issues, so contacted Carl Petersen directly by email to ask if in the Avermann et al., 2012 study, the cells labelled NFS were both SOM and 5HT_3A_R, or just 5HT_3A_R. He replied to say, “We think that our NFS cells are almost exclusively 5HT_3A_R-expressing GABAergic neurons (although this is not proven)”. We then based our model on this assumption.

This assumption that NFS are strictly 5HT_3A_R has led to some perhaps misleading choices in model construction and choices for intrinsic and synaptic parameters. For example, ~40% 5HT_3A_R cells are VIP+ in S1 (Lee et al., 2010), and a primary target of VIP+ interneurons are SOM+ (Pfeffer et al., 2013; Dalezios et al., 2002), but the model has no 5HT_3A_R→ SOM connections. Also, it has 5HT_3A_R→5HT_3A_R connections, but not SOM→PV (Pfeffer et al., 2013)?

We agree that the model could be made more detailed, and of course makes approximations by lumping together all 5HT_3A_R cells into one population. However, splitting the population would involve multiplying the number of model parameters. Designing such models necessarily involves some trade-off between adding more detail versus availability of constraining data, so we settled at some point on this trade-off. A second difficulty with designing such models is the fact that different studies from different labs, or conducted in different brain areas/animal ages, etc often result in mutually inconsistent findings.

Overall, these model choices will likely not affect the main results – an overall varied sensitivity to cellular changes. However, it makes the specifics of how distinct populations are affected in Figure 1 and Figure 2 (and the associated text) difficult to interpret from a biological standpoint. Along those lines, Table 1 describes the circuit model variables and could serve as a resource for modelers and experimentalists alike. However, for clarity, readers should be aware of the assumptions made so that they can use with caution.

This is a fair point and we have now added a paragraph to the Methods making clear these simplifications: “An important caveat is that although this model may be considered detailed by some measures, it also simplifies many aspects of L2/3 circuit. For example, we assumed that all 5HT_3A_R cells were homogeneous, even though they likely separate into different subclasses with type-specific connectivity (Gentet, 2012; Petersen and Crochet, 2013). Layer 2 and layer 3 may also consist of distinct cell populations (Petersen and Crochet, 2013). Not all likely connections were included in the model (Dalezios et al., 2002; Pfeffer et al., 2013), and connectivity was assumed to be random, even though it is likely non-random (Tomm et al., 2014). Although these choices will likely not affect the conclusions of the current study, they may be important to consider for future work that seeks to understand the biological function of the L2/3 somatosensory microcircuit.

4) The authors found that the direction of circuit parameters change in young to mature mice is opposite in KO vs WT, which could not be captured by examining neural activity statistics. Given that there is so much redundancy and variability at the cellular level, how does knowing about these circuit properties help us in terms of mechanisms of change?

We would argue that the redundancy at the cellular level in fact strengthens the case for looking at the circuit level, instead of worrying so much about the neural component changes. We hope that this point is made clearer in the new version of the manuscript.

The logistic response model also fits the threshold s.d., the slope s.d., and the slope-threshold correlation. How were these parameters affected in development/in KO?

We have now added Figure 5—figure supplement 2 that displays all five fitted parameters across genotypes and development. The most obvious changes across development were in the mean slope and threshold. This was why we focused on these in the original manuscript.

Nevertheless, we also found evidence for subtle changes in slope and threshold s.d. between groups, and interestingly also a possible increase in slope-threshold correlation in *Fmr1* KO P30-40 animals. We have added a clarifying note to the results text that points to this supplemental material: “For each animal, we plot the full 5-D parameter fits for all animals in Figure 5—figure supplement 2. We will focus on the most prominent parameter changes which were in mean slope and mean threshold, but we also note an increase in the slope s.d. between P14-16 and P30-40 WT animals that was not observed in *Fmr1* KO, mirroring the increased heterogeneity in firing rates in the same animals (Figure 1).

5) It is unclear what we learn from the population-level P(spike) model that we could not learn from the single neuron model. Two things it would be good for the authors to address here:i) The population-level model lacks motivation I would expect the fourth paragraph of subsection “Firing rates and correlations from the logistic model” to start with something like "In order to characterise the population, we fitted… because…". Why fit a 5-parameter model to the whole population: why not just fit each neuron with a 2-parameter model, and average over their fits? [Just as was done for the model]. Then one would also obtain the population variation in the slope and threshold parameters, and that variation need not conform to a Gaussian model as is enforced by the 5-parameter model. I suspect there is an underlying issue with having enough data per-neuron. Please elaborate.

The reviewer’s suggestion to fit the logistic model for each neuron independently is a good idea, but unfortunately it is a difficult statistical problem. The logistic model has two parameters per neuron (slope and threshold) so fitting it separately for each of N neurons would imply fitting 2N parameters total. However, if the goal was to match the firing rates and pairwise correlations of all the neurons, then we would be faced with N + N*(N-1)/2 constraints, which is much greater than 2N, for all N> 3. Hence the problem would be vastly overconstrained and therefore likely could not be solved exactly. It might be possible to attempt to find the 2N parameters that come “closest” to matching the constraints, but that is in general a difficult problem that we are interested in pursuing but is beyond the scope of this paper. Instead by using the 5-parameter model we flipped the problem to an underconstrained version where we use only 3 target constraints.

ii) Having fitted a population model, not much is done with it. It seems plausible that the variation in slope and threshold (quantified by the SD) could change with development, as the underlying parameter ranges contract or expand. So what is the variation over development, and does it differ between WT and Fmr1 mice?

As discussed above in response to a previous reviewer comment, the changes in s.d. across development/genotypes were non-zero but not as widespread as the changes in mean slope and threshold. To show these to the reader we have now added a new Figure 5—figure supplement 2.